# Transient non-local interactions dominate the dynamics of measles virus N$_{TAIL}$
Lillian Otteson[1,2,7], Gabor Nagy [ID][3,7], John Kunkel[1,2], Gerdenis Kodis [ID][1,2], Lars V. Bock [ID][3], Christophe Bignon[4], Sonia Longhi [ID][4], Wenwei Zheng [ID][5] ✉, Helmut Grubmüller [ID][3] ✉, Andrea C. Vaiana [ID][3,6] ✉ & Sara M. Vaiana [ID][1,2] ✉

The RNA genome of measles virus is encapsidated by the nucleoprotein within a helical nucleocapsid that serves as a template for both transcription and replication. The intrinsically disordered domain of the nucleoprotein (N$_{TAIL}$) is essential for binding the polymerase complex responsible for viral transcription and replication. As for many IDPs, binding of N$_{TAIL}$ occurs through a short molecular recognition element (MoRE) that folds upon binding, with the majority of N$_{TAIL}$ remaining disordered. Although N$_{TAIL}$ regions far from the MoRE influence the binding affinity, interactions between them and the MoRE have not been investigated in depth. Relying on photo-induced electron transfer (PET) experiments between tryptophan and cysteine pairs placed at different positions in the protein under varying salt and pH conditions, combined with analytical models, simulations, and coevolutionary analysis, we identified transient interactions between two disordered regions distant in sequence, which dominate N$_{TAIL}$ dynamics, and regulate the conformational preferences of both the MoRE and the entire N$_{TAIL}$ domain. We propose mechanisms by which these non-local interactions may regulate binding to the measles phosphoprotein, polymerase recruitment, and ultimately viral transcription and replication. Our findings may be extended to other IDPs, where non-local intra-protein interactions affect the conformational preferences of intermolecular binding sites.

The measles virus (MeV) is a non-segmented, single-stranded, negative-sense RNA virus, in which the viral genome is wrapped within a helical nucleocapsid, made of thousands of repeats of the nucleoprotein (N)[1]. This structure is shared among all Paramyxo- and Pneumoviruses, which include common human viruses (e.g., MeV, mumps, respiratory syncytial virus, metapneumovirus, and other para-influenza viruses) and animal viruses (e.g., Sendai virus, Newcastle disease virus and rinderpest virus), as well as zoonotic biosafety level 4 agents of great concern to public health (e.g., Hendra and Nipah viruses, respectively HeV and NiV)[2]. The N protein not only provides the interaction with the RNA and holds the nucleocapsid in its helical structure, but it is also the substrate used for viral transcription and replication. Viral transcription and replication in Paramyxoviruses are ensured by a complex formed by the large-polymerase (L) and the phosphoprotein (P), where the latter enables tethering L to the nucleoprotein-RNA (N-RNA) template for transcription and replication[2,3].

In MeV, N consists of a folded 400 amino acid (a.a.) domain (N$_{CORE}$), which binds the RNA and forms the structured helical nucleocapsid, and of an intrinsically disordered 125 aa. C-terminal domain (N$_{TAIL}$), which protrudes radially outward from the helical nucleocapsid structure[4–7]. The intrinsically disordered N$_{TAIL}$ domain binds to the folded X domain (P$_{XD}$) of the P protein and is essential for MeV transcription and replication. P in turn, binds as a tetramer to both L and N$_{CORE}$[6,8–11]. A similar structure and binding of N$_{TAIL}$ to P$_{XD}$ are found in NiV and HeV[3,12]. The binding of N$_{TAIL}$ to P$_{XD}$ is thought to assist in positioning L appropriately on the nucleocapsid to read the RNA, while allowing efficient transcription re-initiation at intergenic regions[13] and hence progression from one reading frame to the next. It may also assist in maintaining contact between L and the nucleocapsid template, preventing it from leaving before transcribing all the viral genome[11,14]. In addition, the presence of N$_{TAIL}$ loosens the nucleocapsid helical structure, possibly facilitating transcription and replication by L[11].

[1]Center for Biological Physics, Arizona State University, Tempe, AZ, USA. [2]Department of Physics, Arizona State University, Tempe, AZ, USA. [3]Department of Theoretical and Computational Biophysics, Max Planck Institute for Multidisciplinary Sciences, Göttingen, Germany. [4]Aix Marseille Univ, CNRS, AFMB, Marseille, France. [5]College of Integrative Sciences and Arts, Arizona State University, Mesa, AZ, USA. [6]Present address: Department of Chemistry and Chemical Biology, University of New Mexico, Albuquerque, NM, USA. [7]These authors contributed equally: Lillian Otteson, Gabor Nagy. ✉e-mail: wenweizheng@asu.edu; helmut.grubmueller@mpinat.mpg.de; avaiana@unm.edu; sara.vaiana@asu.edu

Because of its crucial role in transcription and replication, binding of MeV $N_{TAIL}$ to $P_{XD}$ has been extensively studied[15–21]. It occurs through coupled folding and binding of a short molecular recognition region (MoRE) of 18 a.a. (gray in Fig. 1)[21]. Upon binding, the MoRE folds into an α-helix, forming a four-helix bundle with three helices from $P_{XD}$ (PDB: 1t6o[8,9]). However, the majority of $N_{TAIL}$ (i.e., the remaining 109 a.a. on the two sides of the MoRE) remains disordered in the bound complex[21]. The $N_{TAIL}$-$P_{XD}$ complex is structurally similar in MeV, NiV and HeV[3,12]. While the regions outside the MoRE (from here on referred to as "flanking regions") do not fold, or engage in stable binding interactions[16,17], they modulate binding affinities, as indicated by the binding affinity increase in truncated variants of $N_{TAIL}$ containing the intact MoRE[22,23]. This increase might be due to a pure excluded volume (entropic) repulsion of the flanking regions, as well as to potential interactions involving the flanking regions of $N_{TAIL}$ (either at the intra- or inter-protein level)[22,24,25].

For $N_{TAIL}$ free in solution, the secondary structure propensity of the MoRE region has been characterized by NMR on full-length $N_{TAIL}$[7], and by all-atom simulation studies of the MoRE fragment[19,26]. In contrast, we know very little about potential interactions between residues that are distant in sequence (non-local interactions), despite their particular importance for $N_{TAIL}$ binding to $P_{XD}$[24,27] and, more generally, for IDP function. Large-scale reconfiguration dynamics can regulate IDP binding mechanisms, rates, and affinities; in fact, examples of this are found even in simple models of proteins undergoing two-state conformational transitions[28]. For $N_{TAIL}$, we expect interactions involving the regions flanking the MoRE on either side to confer binding specificity by dictating large-scale reconfiguration dynamics. In addition, non-local interactions potentially act as allosteric regulators in IDPs by stabilizing certain conformational states. The resulting shift in the population of states within the ensemble can lead to allosteric regulation, as described in the ensemble allosteric model for IDPs[29,30].

Probing non-local intra-chain interactions is therefore essential for a molecular understanding of IDP dynamics, binding, and function[22,31–33]. Direct experimental probing of such interactions is particularly difficult, due to their transient nature, and the fact that IDPs undergo large conformational changes on sub-microsecond time scales. Atomistic simulations of IDPs, on the other hand, are equally challenging due to the wide conformational space sampled by IDPs. To overcome these experimental and theoretical challenges, we leveraged the high sensitivity of IDP dynamics to non-local interactions. We experimentally quantified the dynamics of full-length $N_{TAIL}$ in solution by probing intramolecular contact formation times, using photo-induced electron transfer (PET) between a tryptophan (W) and a cysteine (C) placed in the sequence[34–37]. Using this technique, we measured contact formation rates along the well-defined C-W distance reaction coordinate, probing large-scale backbone motions on the hundreds of nanoseconds to microsecond internal reorganization time scale[36], which is most relevant to IDP function[38]. With minimal perturbation to the sequence, compared to introducing large prosthetic dyes, we achieved a high sensitivity at short spatial distances. To obtain a structural interpretation and a causal description, we compared the measurements to corresponding observables derived from both simulations and analytical models. Our integrated approach allowed us to identify key intra-protein interactions

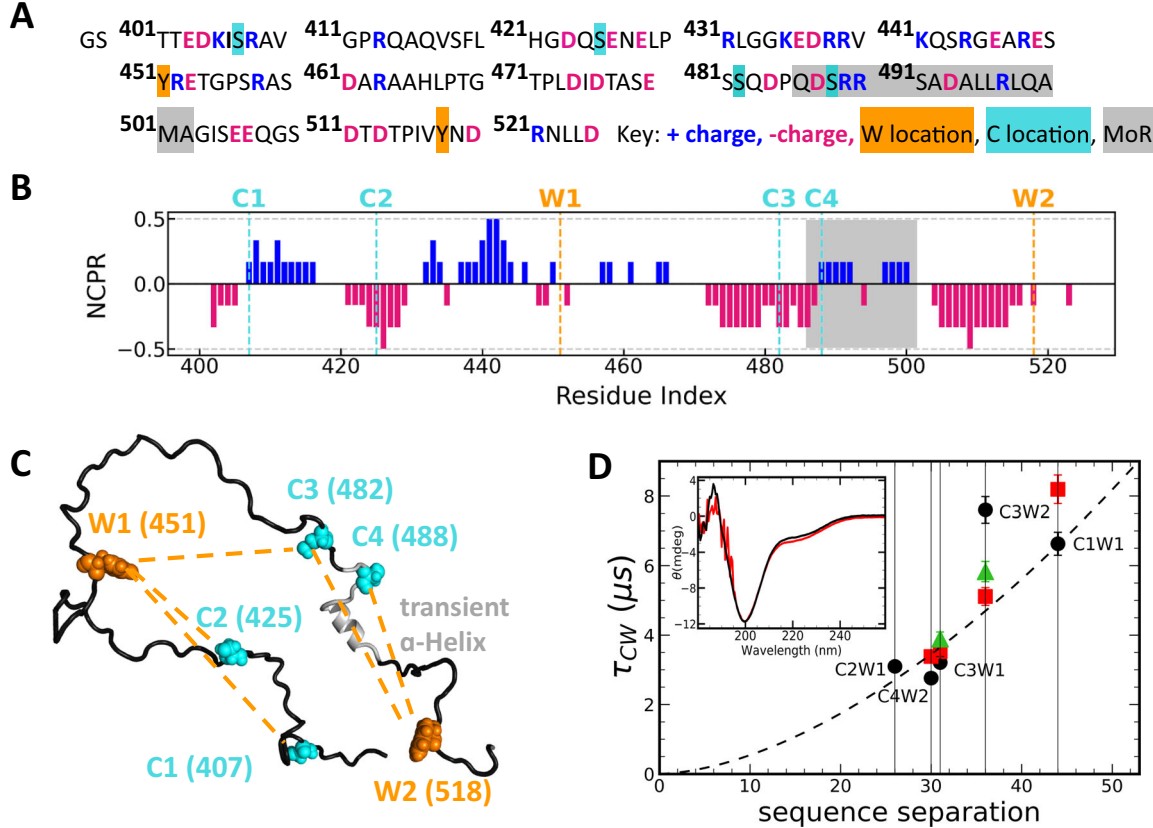

**Fig. 1 | Results of photo-induced electron transfer (PET) experiments at** $T = 20\ °C$, **on** $N_{TAIL}$ **variants containing a unique C-W pair. A** $N_{TAIL}$ sequence, showing charged residues at pH 7.6, locations of Y to W and S to C substitutions, and location of the MoRE, which folds into a helix upon binding to $P_{XD}$. **B** Net charge per residue calculated using a window size of six amino acids. **C** W and C positions. Each variant contains a unique C-W pair. The C-W distance in each variant is indicated by orange dashed lines. C3W2 and C4W2 variants encompass the MoRE transient helix. **D** C-W relaxation time $\tau_{CW}$ due to PET as a function of C-W sequence separation. Experimental conditions: pH 7.6 and 150 mM NaCl (black circles); pH 4.0 and 150 mM NaCl (red squares); pH 7.6 and 500 mM NaCl (green triangles). Dashed line: expected scaling of $\tau_{CW}$, from a SAW-ν homopolymer model at pH 7.6 (black). Inset: Comparison of $N_{TAIL}$ circular dichroism spectra at pH 7.6 (black) and pH 4 (red).

that extend beyond the MoRE region, and play an important role in $N_{TAIL}$ dynamics and likely also in function.

## Results

### PET reveals the heteropolymeric nature of $N_{TAIL}$ dynamics

To characterize the dynamics of $N_{TAIL}$ in solution, we probed intramolecular contact formation, using PET between tryptophan (W) and cysteine (C) residues at different positions in the sequence[35–37]. After exciting W to the triplet state using a nanosecond UV laser pulse, we monitored the triplet state population in time via transient absorption measurements. As $N_{TAIL}$ undergoes spontaneous conformational changes, W can come into contact with C, and PET can occur. During PET, the excited-state (triplet) electron is transferred from W to C, forming a W radical cation, which eventually relaxes to the ground state. The relaxation time due to PET, $\tau_{CW}$, therefore provides information on how often and quickly C and W come into contact after W excitation. If the C-W contact formation rate is of the same order of magnitude or faster than the natural decay rate of W, $\tau_{CW}$ can be obtained from the short-time behavior of the triplet state relaxation curves (Supporting Fig. S1).

The wild-type $N_{TAIL}$ sequence (Fig. 1A) does not contain native W or C residues. To probe the $N_{TAIL}$ conformational dynamics, we generated five $N_{TAIL}$ variants (PET variants), each bearing a unique C-W pair, to probe the five distances indicated in Fig. 1C. To minimize perturbation of the chemical properties, we substituted native tyrosine (Y) residues with W, and native serines (S) with C residues. To determine the natural decay rate of excited W at the two positions, we also generated two variants with the respective W substitution, but lacking any C substitution. The Supporting Information contains the sequences of all seven $N_{TAIL}$ variants (Table S1) and details about their production and purification (Supporting Methods S1).

Relaxation times $\tau_{CW}$ were estimated from multiple, independent sample preparations for each $N_{TAIL}$ variant, and a series of repeated measurements to account for W and C photodamage during the measurements (Supporting Methods S2). All measured relaxation curves and fitted functions to obtain $\tau_{CW}$ are shown in Figs. S3–S16, with the fit results summarized in Table S2.

We first performed measurements with the five PET variants of $N_{TAIL}$ under near-physiological conditions (i.e., 15 mM Tris, 150 mM NaCl, 1 mM TCEP, pH 7.6, in the following referred to as "reference conditions"). Figure 1D shows the $\tau_{CW}$ (black circles) estimated from these measurements as a function of the W-C sequence separation ($|i-j|$).

To test whether the estimated $\tau_{CW}$ can be explained based on C-W sequence separation alone, we compared them to relaxation times expected from a simple homopolymer model (Fig. 1D, dashed line), where the only non-local interactions between the residues are excluded volume interactions. In this model, both the physical distance in space between C and W at positions $i$ and $j$, as well as the corresponding relaxation time $\tau_{ij}$, scale with the sequence separation $|i-j|$ [39]. The scaling relation for relaxation times $\tau_{ij} = \tau_r |i - j|^{\nu_\tau}$ depends on a prefactor $\tau_r$ and an exponent $\nu_\tau$. The exponent $\nu_\tau = 1.69$ (corresponding to a Flory scaling exponent $\nu$ of 0.52) was estimated from small-angle X-ray scattering (SAXS) measurements of $N_{TAIL}$ [6] based on a homopolymer model (SAW-$\nu$ model[40], Supporting Methods S4). The prefactor $\tau_r = 0.011$ μs was obtained by fitting to the estimated $\tau_{CW}$ (Fig. 1D) from PET experiments.

The general trend of the estimated $\tau_{CW}$ at physiological pH (Fig. 1D, black circles) follows the simple homopolymer scaling (dashed curve); however, some contacts form faster and others slower than expected. This dynamic heterogeneity (deviation from the homopolymer model) observed along the sequence is most striking when comparing the C3W2 ($|i-j| = 36$) and C4W2 ($|i-j| = 30$) variants for which the C-W pairs encompass the MoRE region. These variants share the W position and differ by just six residues in sequence separation between W and C (Fig. 1A–C). According to the homopolymer model, such a difference in sequence separation would result in a $\tau_{CW}$ increase by a factor of 1.3. In contrast, the $\tau_{CW}$ estimated from relaxation curves increases by a factor of 2.8. This considerable deviation from the homopolymer behavior is primarily due to the measured

7.6 μs relaxation time for C3W2, which is substantially longer than the 4.7 μs homopolymer estimate. The fact that this anomalously slow relaxation time is not reproduced by the homopolymer model indicates that the $N_{TAIL}$ dynamics near the MoRE region are considerably affected by more complex heteropolymeric interactions, likely secondary structure formation, electrostatic interactions between charged residues, or solvent-induced interactions.

### Charged amino acids partially account for $N_{TAIL}$ dynamic heterogeneity

To test whether interactions between charged residues or secondary structure formation cause the dynamic heterogeneity, we performed additional experiments at low pH and increased salt concentrations, i.e. under conditions that should modulate electrostatic interactions.

The $N_{TAIL}$ sequence contains 40 charged residues (13 Asp, 10 Glu, 14 Arg and 3 Lys), often clustered together (Fig. 1B). If electrostatic interactions between charged residues were markedly contributing to the observed dynamic heterogeneity, we would expect that screening the interactions by increasing the salt concentration would decrease or eliminate the heterogeneity. Indeed, increasing the salt concentration from 150 mM (Fig. 1D, black circles) to 500 mM (green triangles) results in a $\tau_{CW}$ decrease from 7.6 μs to 5.8 μs for C3W2, and an increase from 3.2 μs to 3.9 μs for C3W1, bringing both data points close to the homopolymer curve. This suggests that electrostatic interactions are a major contributor to the observed dynamic heterogeneity of $N_{TAIL}$.

Next, we investigated why electrostatic interactions accelerate relaxation for the C3W1 variant, but slow down relaxation for the C3W2 variant. We hypothesized that changes in contact dynamics are caused by the non-uniform distribution of charged amino acids (charge patterning[41,42]) along the $N_{TAIL}$ sequence. However, the increased electrostatic screening at higher salt concentrations could also affect the stability of the transient helix in the MoRE region, and thereby modulate the C3W2 relaxation time. To address these two possibilities, we performed coarse-grained molecular dynamics (CG) simulations that allowed us to explicitly include both charge patterning and secondary structure contributions.

For these simulations, we used a modified version of the recently developed residue-based HPS model[43] for IDPs. The CG model was adjusted to reproduce the experimental radius of gyration of $N_{TAIL}$ obtained from SAXS[6] and the helical propensity of the MoRE region from NMR[7]. The effects of varying salt concentrations in the CG model were described by Debye–Hückel electrostatic screening[44]. Additional details on the CG simulations are provided in Supporting Methods S5.

To determine the effect of increased salt concentration (150 mM to 500 mM), we computed the mean distance for each pair of amino acids $i$ and $j$ from the CG simulations at both salt concentrations. Figure 2A shows the difference between these means, highlighting the compaction/expansion of each $N_{TAIL}$ segment. In the simulations, the N-terminal part of $N_{TAIL}$ up to C3 expands at higher salt concentrations (red areas), whereas the C-terminal part becomes more compact (blue areas).

The positively and negatively charged amino acids are well-blended in the N-terminal region, with a close to zero net charge. The N-terminus, therefore, behaves like a polyampholyte, where interactions between oppositely charged residues result in compact conformations and screening these electrostatic interactions at higher salt concentrations results in more expanded conformations. This expansion, in turn, increases the average distance between C3 and W1, leading to a slower expected relaxation time at high salt concentration. In contrast, the C-terminal region has more negatively charged amino acids than positively charged amino acids under physiological conditions. This imbalance leads to net repulsive electrostatic interactions that keep this region expanded. Increasing the electrostatic screening at high salt concentrations reduces the repulsion, which leads to the compaction of the C-terminus of $N_{TAIL}$, a decrease in the mean C3W2 distance, and an expected decrease in the C3W2 relaxation time.

To determine if the salt-induced changes in relaxation times can be fully described by changes in inter-residue distances due to charge

**Fig. 2 | Coarse-grained (CG) simulation of $N_{TAIL}$.** Differences between the mean pairwise distances ($R_{i,j}$) of the CG simulations at the reference condition (0.15 M salt and pH 7.6) and simulations (**A**) at high salt concentration (0.5 M salt, pH 7.6) or **C** low pH (0.15 M salt pH 4.0), respectively. The diagonals show the charged amino-acid positions with positively charged amino acids in blue and negatively charged amino acids in magenta. The MoRE is highlighted in gray. The full amino acid sequence is provided in Fig. 1A. **B** Ratios of C-W relaxation times due to PET, calculated from CG simulations (bars) at different conditions (high salt: green, or low pH: red), relative to the reference condition. For comparison, experimental ratios are shown where available (black squares). **D** C-W relaxation times for all five $N_{TAIL}$ PET variants estimated from CG simulations at different conditions (reference: black, high salt: green, low pH: red). The blue triangles denote the relaxation times for the C3W2 and C4W2 variants after varying the MoRE helix content (helix-35%, helix +: 60%). All CG relaxation times were scaled by a factor chosen so that the same homo-polymer fitting function (dashed line) of Fig. 1D fit both experimental and CG data.

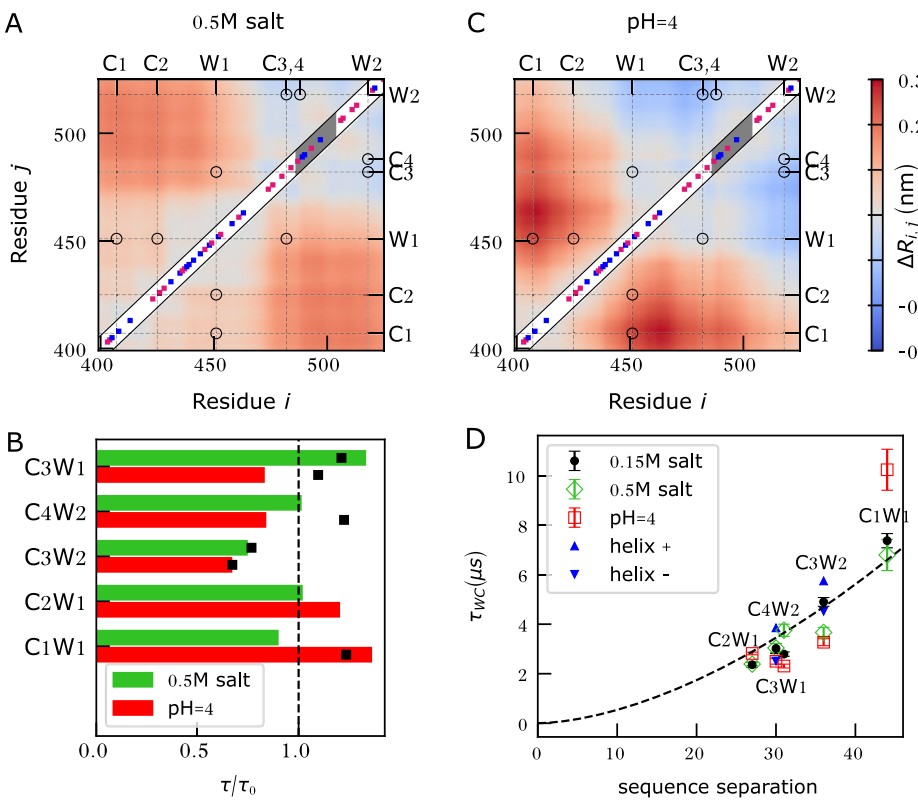

patterning alone, we estimated the expected ratio of relaxation times ($\tau/\tau_0$) from the CG simulations at 500 mM and 150 mM NaCl concentrations, respectively. For these calculations, we assumed reaction-limited quenching kinetics, in which case, the relaxation time is inversely proportional to the probability of W and C being at a reactive distance. We used a cutoff distance of 1 nm to determine the probability of reactive conformations (further details in Supporting Methods S5). Figure 2B shows the $\tau/\tau_0$ ratios upon increasing the salt concentration for each variant (green bars). A ratio larger than one indicates an increase in the relaxation time, exclusively due to the expansion of that segment. Comparison with experimental ratios (Fig. 2B, black squares) shows that the CG simulations accurately predict both the increase in C3W1 relaxation times and the decrease in C3W2 relaxation times, which indicates that charge patterning suffices to explain the salt-induced changes in $N_{TAIL}$ relaxation times. It also corroborates the hypothesis that electrostatic interactions between charged residues are a key factor in the observed dynamic heterogeneity in $N_{TAIL}$.

To further test the effect of electrostatic interactions on the dynamic heterogeneity of $N_{TAIL}$, we measured C-W relaxation times upon decreasing the pH from 7.6 to 4.0, while maintaining a near-physiological salt concentration (150 mM) (Fig. 1D, red squares). In contrast to increasing the salt concentration, which screens all electrostatic interactions, decreasing the pH only alters the charge on specific residues that titrate in the 7.6–4.0 pH range, specifically negatively charged Asp and Glu may become neutral, and neutral His may become positively charged. If electrostatic interactions involving these charged residues were the main contributor to the dynamic heterogeneity observed at physiological pH, we would expect a reduction in heterogeneity when decreasing the pH to 4.0. In particular, we expect the N-terminal part, which acts as a balanced polyampholyte at pH 7.6, to become positively charged and therefore to expand at pH 4.0, even beyond what was observed at high salt concentrations. This expansion should then increase the PET relaxation times for the C1W1 and C2W1 variants. Conversely, we expect the negatively charged C-terminal part to become more neutral, resulting in more compact conformations that reduce the C3W2 and C4W2 relaxation times.

To capture the effects of lower pH in CG simulations, the charges of the residues that titrate between 7.6 and 4.0 were adjusted to match the average level of protonation at the simulated pH. To that aim, the charges assigned at pH 7.6 (−1.0 for Asp and Glu and +0 for His residues) were changed to partial charges at pH 4.0 (−0.69 for Asp, −0.36 for Glu and +0.99 for His). Further details on charge assignments are provided in Supporting Methods S5. If electrostatic interactions involving specific charged residues drive the dynamic heterogeneity at physiological pH, lowering the pH to 4.0 should reduce this heterogeneity. The N-terminal region, a balanced polyampholyte at pH 7.6, is expected to become positively charged and expand, while the negatively charged C-terminal region should become more neutral and compact. The CG simulations with simple fixed-charge adjustment at low pH are consistent with the expected behaviors described above, i.e., the N-terminal region expands and the C-terminal region becomes more compact (Fig. 2C), which is also reflected in the calculated relative relaxation times (red bars Fig. 2B).

The C-W relaxation times of all five PET variants estimated from measurements at pH 4.0 are closer to the homopolymer curve (Fig. 1D, red squares) than those at pH 7.6. Lowering the pH also reduces the observed dynamic heterogeneity between the C3W2 and C4W2 variants, with the ratio of relaxation times reduced from 2.8 at pH 7.6 to 1.5 at pH 4.0, which is closer to the homopolymer prediction of 1.3. However, in contrast to the reduction of the C-W relaxation time expected from compaction based on charge patterning and predicted by CG simulation, the measured relaxation time for C4W2 increases slightly upon lowering the pH (Fig. 1D).

In Fig. 2B, the ratios of relaxation times at pH 4.0 and pH 7.6 estimated from the CG simulations (red bars) were compared to those estimated from PET experiments (black squares). The CG model correctly captures the trend of C1W1 and C3W2, but not that of C4W2 and C3W1. Further, the measured increase of $\tau_{CW}$ for C4W2 is not only contrary to the CG simulation results, but also contrary to the expectation based on charge patterning, suggesting that other unaccounted factors influence the relaxation times. In the following, we will consider three factors that can affect relaxation times in a pH-dependent manner: the MoRE helicity, W to C

electron transfer efficiency, and residue-specific changes in protonation equilibria.

**pH dependence of MoRE helix content**. The CG model suggests a limited difference in helical propensity upon changing pH (Supporting Fig. S20). We further measured and compared the far-ultraviolet circular dichroism (CD) spectra of the $N_{TAIL}$ variant Y518W (W2) at pH 7.6 (Fig. 1C, inset, black curve) and 4.0 (red curve). CD spectroscopy in this wavelength range of 180–260 nm is highly sensitive to the presence of α-helices, and the two spectra are very similar, indicating that the helical content of $N_{TAIL}$ is not affected significantly by pH. In addition, analysis of the measured spectra by the SESCA software[45] (Supporting Methods S13) predicts at most 8% global helix content, which is compatible with the NMR estimates used to set the MoRE helicity in CG simulations. Together, these results suggest that the MoRE helix content is not affected by the change in pH.

**Changes in electron transfer efficiency**. Because PET involves the formation of W radical cations, the pH may directly affect the electron transfer process in solution, independent of pH-induced changes in the protein conformational ensemble and dynamics. To address this possibility, we performed a bimolecular quenching study between N-acetyl-tryptophan-amide (NATA) and cysteine molecules, freely diffusing in solution, at pH 7.6 and 4.0 (Supporting Methods S2). In these experiments, the bimolecular quenching rate was determined by measuring the W triplet relaxation times in samples at fixed NATA concentration and increasing cysteine concentrations. These measurements showed that the C-W bimolecular quenching rate (determined from the slopes of Stern-Vollmer plots) at pH 7.6 and 4.0 is very similar (Fig. S19). This result shows that the efficiency of electron transfer from the excited state W to C, under the conditions of our PET experiments, is not directly affected by pH, and the pH dependence of $\tau_{CW}$ in $N_{TAIL}$ is due to changes in the protein structure and dynamics.

**Residue-specific protonation**. $N_{TAIL}$ is rich in Asp and Glu, amino acids that reach protonation equilibrium close to pH 4.0. Under these conditions, the protonation state of individual acidic residues can be influenced, e.g., by the spatial vicinity of similarly or oppositely charged residues, resulting in a dynamic, conformation-dependent charge distribution[46]. This charge distribution would also allow electrostatic interactions to stabilise certain conformational states, increasing and decreasing the observed $\tau_{CW}$ for individual $N_{TAIL}$ variants compared to those predicted by our fixed-charge CG simulations.

**Dynamic heterogeneity at the $N_{TAIL}$ C-terminus**
Although dynamic protonation may explain deviations from CG and homopolymer models, this effect is an unlikely explanation for the dynamic heterogeneity between C3W2 and C4W2 under reference conditions at pH 7.6. To address this heterogeneity with CG simulations, we assumed these processes are reaction-limited, and that $\tau_p$, the proportionality between C-W contact probabilities ($P_{r_{ij} < r_0}$) and corresponding $\tau_{CW}$ relaxation times, is the same for all PET variants, and thus $\tau_{ij} = \tau_p / P_{r_{ij} < r_0}$. Under these assumptions, it is possible to compare relaxation times of different C-W pairs obtained from CG simulations relative to one another and address the dynamic heterogeneity at the $N_{TAIL}$ C-terminus.

To this aim, we fitted a homopolymer model to these CG-derived relaxation times at the reference condition, and chose $\tau_p$ such that the scaling equation is identical to the one obtained from experimental measurements in Fig. 1D. The resulting relaxation times from CG simulations for all three conditions (reference, high salt, and low pH) are shown in Fig. 2D. The figure shows that all CG-derived relaxation times at the reference condition (black circles) follow homopolymer scaling (dashed black line), with only small deviations for individual CW pairs. Although the CG simulations accurately predict many of the changes upon increasing salt concentrations or lowering the pH, they do not show either the considerable

slowdown for the C3W2 relaxation time, or the pronounced dynamic heterogeneity measured at pH 7.6 between C3W2 and C4W2 (ratio 2.8). In fact, the CG-derived $\tau_{C3W2}/\tau_{C4W2}$ ratio of 1.6 is close to the 1.3 ratio predicted by homopolymer models, suggesting that the CG simulations do not capture the main cause of the dynamic heterogeneity observed in PET measurements.

An additional reason why the CG simulations may not have captured the C3W2/C4W2 dynamic heterogeneity is that we underestimated or overestimated the MoRE helix propensity (~45%) under the conditions of PET experiments. This can alter the CG-derived heterogeneity if the MoRE helix in its folded state affects C3W2 and C4W2 relaxation differently. To test this hypothesis, we performed additional CG simulations in which we altered dihedral potentials that control the MoRE helicity and computed the $\tau_{CW}$ for C3W2 and C4W2 variants at high (60%) and low (35%) helix contents (blue triangles on Fig. 2D). Relaxation times from these simulations show that considerable changes in the helix content can indeed shift the relaxation times obtained from CG simulations, but this shift affects both C3W2 and C4W2 similarly. Therefore, even a significant error in the imposed MoRE helicity would have only a small ( ± 0.1) effect on $\tau_{C3W2}/\tau_{C4W2}$ ratio, certainly not sufficient to explain the dynamic heterogeneity observed in PET experiments.

Overall, our results show that $N_{TAIL}$ dynamics are heteropolymeric in nature, especially in the region encompassing the MoRE, with electrostatic interactions playing an important role. It appears that the dynamic heterogeneity of $N_{TAIL}$ cannot be explained by a simple polymer scaling model (Fig. 1D) or by CG models including helix formation within the MoRE and explicit interactions between charged amino acids (Fig. 2). These observations suggest that the cause of the dynamic heterogeneity is either side-chain specific transient interactions, such as salt bridges and hydrogen bonds[47], or interactions between the protein and solvent. To study the effects of these interactions on $N_{TAIL}$ dynamics in more detail, we turned to all-atom explicit-solvent molecular dynamics (MDs) simulations.

**Disordered $N_{TAIL}$ is characterized by a conformational ensemble with distinct states**
In addition to specific interactions between amino acids and protein-solvent interactions, all-atom simulations include the backbone and side-chain dynamics required to accurately compute the contact formation rates. The relaxation times $\tau_{CW}$ estimated from these contact formation rates can be compared to $\tau_{CW}$ derived from PET experiments. However, this increased level of detail comes with two drawbacks; considerably higher computational costs that limit conformational sampling and the need for a larger number of accurate interaction parameters (force field). IDP simulations are particularly sensitive to variations in the applied force field[48–51], and force field accuracy for IDPs often varies in a system-dependent manner[52].

To maximize conformational sampling, we limited all-atom simulations to the reference conditions (pH 7.6 at 150 mM NaCl concentration) and focused on the C3W2 and C4W2 variants, which showed large dynamic heterogeneity (Fig. 1D). Further, to ensure the reliability of all-atom simulations, we first validated eight force fields, characterized by different combinations of protein and water parameters, against independent experimental SAXS, CD, and NMR[6,16] data on the wild-type (WT) $N_{TAIL}$ sequence (shown in Supporting Table S3). To evaluate these force fields, we first performed simulations of WT $N_{TAIL}$ using each force field, and then compared the resulting trajectories to the SAXS data, which reports on protein compactness, using the software CRYSOL[53]. Next, we compared the trajectories to CD and NMR chemical shift measurements, which report on secondary structure propensities, using SESCA[45], and SPARTA + [54]. For further details of the force field validation, see Supporting Methods S8.

The two force fields that reproduced the WT validation data most accurately were the CHARMM36m (C36M)[51] force field with optimal point-charge (OPC) water model[55], and Amber99SB-disp (A99SB-d) force field[49]. Both of these force fields produced extended $N_{TAIL}$ ensembles with transiently forming helices in the MoRE region, which was in good agreement with the available experimental data (Supporting Table S3), and

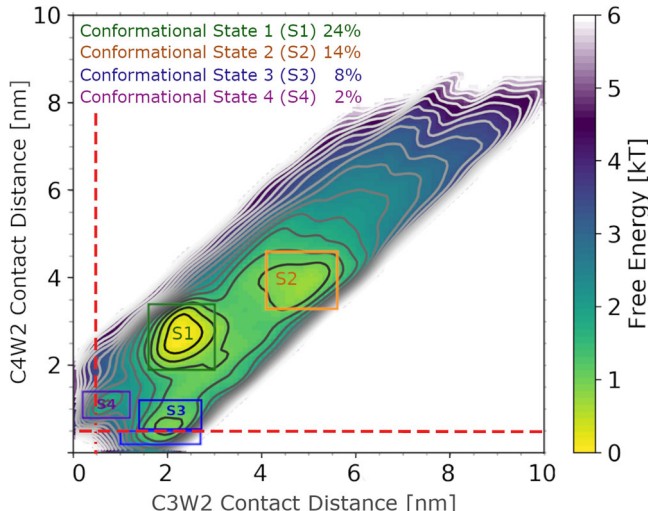

**Fig. 3 | Conformational states revealed by all-atom MD simulations.** The free-energy landscape of sampled $N_{TAIL}$ conformations along the calculated C-W distances from C36M-OPC MD simulations. Contour lines denote a 0.3 kT difference in the estimated free-energy landscape. Conformational states are highlighted by colored labels and rectangles. The population of conformations within each rectangle is shown in the legend. Red dashed lines indicate a contact distance of 0.4 nm. Conformations below these cutoffs would allow quenching of the W triplet state, due to electron transfer to C, for the corresponding variant (C4W2 or C3W2).

suggested helix propensities similar to those reported in computational studies of MoRE region[19,26] (Supporting Methods S23).

The two best force fields were used to perform longer all-atom simulations of the C3W2 and C4W2 variants. As a second force field validation step (Supporting Table S4), the variant trajectories were tested against PET relaxation measurements of the same variants, and the CD spectrum of the W2 variant (Fig. S21B). Simulations of both force fields estimated C3W2 and C4W2 relaxation times, showing considerable dynamic heterogeneity. However, the C36M-OPC force field was the only parameter set that reproduced measurements of both WT $N_{TAIL}$ and the two variants within the experimental and computational uncertainty, including the two measured W relaxation curves (Supporting Fig. S22). Thus, C36M-OPC trajectories were used for all subsequent analysis. The relaxation times $\tau_{CW}$ estimated from these simulations were $2.0 \pm 0.5$ μs for C4W2 and $5.3 \pm 3.3$ μs for C3W2, in good agreement with the corresponding measured values of $2.3$ μs and $7.6$ μs. Although the uncertainty on the C3W2 relaxation time is large due to limited sampling, the ratio of the relaxation times estimated from the simulations is $2.7 \pm 1.8$ close to the experimentally determined ratio of 2.8. Together, these results suggest that the all-atom simulations accurately describe the $N_{TAIL}$ dynamics causing the C3W2/C4W2 dynamic heterogeneity.

To study the origin of the dynamic heterogeneity in detail, we computed the free-energy landscape of the $N_{TAIL}$ variants along the C-W distances for positions C3W2 and C4W2. Variants C3W2 and C4W2 only differ in the exchange of an oxygen and a sulfur atom between two nearby residues at position 482 (C3) and 488 (C4), respectively. We therefore assumed that the free-energy landscape of the two variants is similar and combined their trajectories for computing it. The C-W distances were defined as the distance between the gamma sulfur of C (or gamma oxygen of S) and the nearest heavy atom of the W side-chain ring. These serve as intuitive reaction coordinates for describing C-W contact dynamics in the $N_{TAIL}$ C-terminal region.

Unexpectedly, the computed free-energy landscape (Fig. 3) is characterized by four local minima corresponding to four $N_{TAIL}$ conformational states (S1–S4). Approximately 60% of the sampled $N_{TAIL}$ conformations are within the respective energy wells, including almost all conformations with C-W contacts necessary for PET. S1 corresponds to the conformational state

around the absolute free-energy minimum and is occupied for ~24% of the simulation time. S2 and S3 are less populated conformational states, occupied for 14% and 8% of the simulation time, respectively. S4 is the least populated state, with ~2% of conformations. The landscape shows only small barriers (<1 kT) separating S1, S2, and S3, and a low free-energy barrier between S1 and S4 (<2.0 kT). This shallow and accessible free-energy landscape suggests nearly-free diffusion between states, which is in line with previous SAXS results suggesting $N_{TAIL}$ to be a pre-molten globule with residual structure[6].

The four conformational states influence C-W relaxation times as well. PET cannot occur in the two most populated states (S1 and S2), because the distances between electron acceptors C4 or C3 and electron donor W2 are more than 2 nm. In state S3, C4W2 distances below 0.4 nm are observed, allowing PET in the C4W2 variant, while in state S4, C3W2 distances below 0.4 nm are reached, which allows PET in the C3W2 variant. Further, from the most populated S1 state, the free-energy barrier that $N_{TAIL}$ has to cross to reach S4 is ~0.9 kT larger than the barrier to reach S3. This suggests that S3 is more easily accessible than S4, with an ~2.5-fold difference between the respective transition rates. This result is consistent with and provides a possible explanation for the measured 2.8 ratio between the C3W2 and C4W2 relaxation times.

## Non-local $N_{TAIL}$ interactions explain dynamic heterogeneity

To determine the molecular basis of the observed dynamic heterogeneity, we investigated which specific interactions stabilize the four conformational states identified above. To this aim, Fig. 4 shows intramolecular contact probability maps for each conformational state, with backbone ($C_\alpha$) contacts shown in yellow to green. Because both the pH and salt concentration dependence of PET experiments and CG simulations indicated an important role of electrostatic interactions in $N_{TAIL}$ dynamics, we also calculated contact probability maps for salt bridges between charged amino acids shown in purple to blue. For the sake of clarity, when discussing the contacts, we defined five regions along the $N_{TAIL}$ sequence (Fig. 4 bottom, color-coded and labeled A-E). Regions A-C are subdivisions of the polyampholyte N-terminal part identified in our CG simulations, regions D and E constitute the negatively charged C-terminal part, with the MoRE located in region D.

The contact maps (Fig. 4, right) clearly show that the four states are stabilized by distinct sets of local (within a region) and non-local (between regions) interactions, although some interactions occur in several conformational states. The maps also provide information on the secondary and tertiary structure preferences of $N_{TAIL}$ in these conformational states. Helical secondary structure elements and tight turns appear on contact maps as continuous sets of interactions, parallel and close (within five residues) to the diagonal. β-strands, β-hairpins or longer loops appear as sets of interactions that are further away and either parallel or perpendicular to the diagonal. Disordered regions typically have few or sporadic interactions.

The four conformational states show markedly different structures (Fig. 4, left), including the MoRE in region D. In state S1, region D is largely non-helical but forms several transient local salt bridges. Several non-local interactions (encircled in green) may prevent donor-acceptor contacts for both variants (C3W2 and C4W2). S1 conformations are also stabilized by salt bridges formed between regions B and E. Salt bridges involving K441, R444 (region B) and E507 and D511 (region E) are particularly frequent and probably play an important role in stabilizing this conformational state.

State S2 includes conformations where region D is non-helical and conformations with a folded α-helix (residues 492–502). The helical structures appear on the contact map as an interaction patch close to the diagonal of region D (encircled in orange). Non-local interactions in S2 appear to be less stable than in S1 and include contacts between regions A and C, C and D, as well as B and D. Stable interactions in S2 also involve two salt bridges between R413 and D476 (regions A and C) and K441 and E449 (local in region B). Many sporadic interactions (both backbone contacts and salt bridges) were observed within the N-terminus (between regions A and B). Taken together, the interactions suggest that S2 consists of multiple

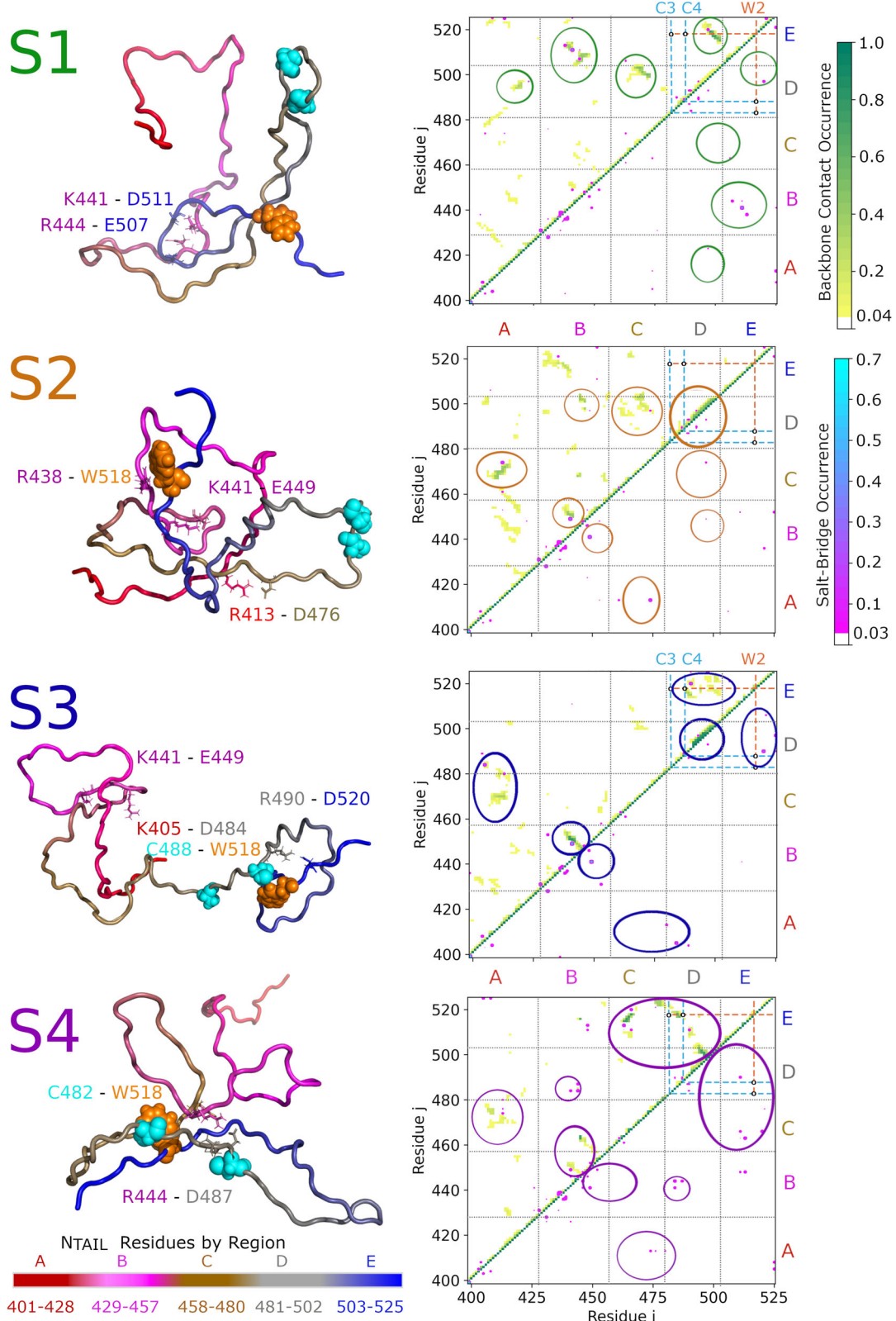

conformational substates and most interactions link central regions (B and C) with both termini (A, D, and E).

Compared to S1 and S2, the conformations of S3 comprise fewer non-local interactions and are instead characterized by two very stable local interaction sites. These local interactions correspond to a shorter helix between residues 492–497 present in all S3 conformations and to a stable

hairpin or zipper in region B, stabilized by a salt bridge (K441-E449) observed previously in S2. Further, there are two sets of notable non-local interactions. First, a patch of interactions between regions D and E folds the C-terminus into a loop, stabilized by the salt bridge between R490 and D520. These interactions also bring the electron donor W2 (518) very close to the potential acceptor at position C4 (488), thus promoting electron transfer in

**Fig. 4 | Contact maps and representative conformations in $N_{TAIL}$ conformational states.** For each conformational state (S1–S4, shown in Fig. 3), a representative conformation (left) and contact probability map (right) is shown. Each map shows $C_\alpha$ contact probabilities (above the diagonal, yellow to green) and salt bridge probabilities (above and below the diagonal, purple to blue). Cysteine acceptor positions C3 and C4 (residues 482 and 488) as well as tryptophan donor position W2 (residue 518) are indicated by cyan and orange dashed lines, respectively. Potential C-W contacts are indicated by black circles. The boundaries of identified interaction regions A–E and their colour-coding for representative conformations are shown in the bottom-left side of the figure. The same regions are separated by black dotted lines on the contact maps. The most prominent interactions in each state are encircled. The pixel size of salt-bridge interactions was increased for visibility, using a Bessel interpolation scheme. On the left side, the backbone trace of a representative conformation from each state is shown in cartoon representation, colored according to regions. The donor and acceptor atoms are highlighted as orange and cyan spheres, respectively. Residues with prominent interactions in the conformational states are labeled and shown in stick representation. Individual contact maps in high resolution, as well as the average contact map for all $N_{TAIL}$ simulations, are shown in Figs. S23–S27.

the C4W2 variant. At the same time, the zipper in region B brings the N-terminal close to regions C and D, where a salt bridge between K405 and D484 may prevent electron transfer to position C3 (482) in the C3W2 variant.

The S4 state allows electron transfer to position C3 (482) in the C3W2 variant. It is structurally very different from states S1–S3. The most defining structural feature of this state is a long loop reminiscent of an anti-parallel β-sheet stretching over the entire C-terminus (regions D and E). This loop is stabilized by interactions between regions C and E, including a salt bridge between R463 and D513. The N-terminus of $N_{TAIL}$ in this conformational state has only weaker interactions, mostly limited to contacts between regions A and C as well as between regions B and C.

To determine if frequent contacts observed in one of the four conformational states (Fig. 4) are specific to that state and if they are correlated with other contacts, including C-W contact formation for the C3W2 and C4W2 variants, we applied normalized pointwise mutual information analysis (NPMI, see Supporting Methods S22).

The NPMI analysis revealed that many salt bridges between regions B, D, and E occur selectively in their respective conformational states. These stable salt bridges often co-occur with clusters of backbone contacts (encircled in Fig. 4), possibly nucleating the interaction clusters that stabilize the states.

Further, all frequent contacts observed in the S1 state were anti-correlated with C-W contact formation (Table S5), indicating that the specific interactions that stabilize the S1 state would increase the expected relaxation times compared to the flat free-energy landscape of a homopolymer model. In contrast, all frequent contacts of states S3 and S4 are positively correlated with C-W contact formation for variants C4W2 and C3W2, respectively. Thus, interactions in these two states selectively reduce the relaxation times of either C4W2 or C3W2 variants, contributing to the dynamic heterogeneity observed in PET experiments.

Finally, the same charged residues (e.g., K441, R497, and D501) in regions B, D, and E form salt bridges with different residues in different conformational states (Table S5). Switching between these sets of interactions is necessary to transition between the most populated conformational states, and therefore likely define the free-energy barriers between the states (Fig. 3). Further, a conformational switch based on these competing intramolecular $N_{TAIL}$ interactions of region B may have a functional role, by regulating the helical propensity of the MoRE in region D, which in turn modulates affinity for the measles phosphoprotein.

### Functional relevance of non-local interactions is supported by co-evolutionary analysis

Having identified the most important interactions that dictate $N_{TAIL}$ structure and dynamics (Fig. 4), we asked whether they also have a functional role. To this aim, we performed a co-evolutionary analysis of $N_{TAIL}$ using EVcouplings[56], which identifies and quantifies correlated mutations of residue pairs within the sequence of homologous proteins from different species or strains, and which often indicates pairwise interactions of functional importance. A total of 12,072 sequences similar to $N_{TAIL}$ have been aligned and selected for this analysis (Supporting Methods S21).

Most non-local residue pairs with positive EVcouplings scores connect regions B and D (Fig. S28), suggesting evolutionarily conserved interactions between these two regions. Here we specifically highlight the four non-local residue pairs with the highest EVcoupling scores (Fig. S28, top left, Table S5). These pairs include hydrophobic contacts involving three leucine residues (L491, L496 and L498) interacting with S443 and A447, and one salt bridge between R444 and D493. Correlated mutations of residues in the distant regions B and D suggest a biologically relevant, functional interaction between these two regions of $N_{TAIL}$. Given that both L491 and L498 are located in the MoRE region (in region D) and are directly involved in binding to $P_{XD}$, it is possible that their interactions with residues of region B serve the function of regulating binding. We note that the contact analysis of our simulations does not identify these specific coevolving residue pairs as frequent interactions. However, all-atom simulations independently identify non-local interactions between regions B and D (Fig. 4, S2 state), including A447 contacts with $P_{XD}$ binding residues L498 and M501. These differences between all-atom simulations and the co-evolution analysis may be due to insufficient sampling or the lack of biological context in all-atom simulations (i.e., the absence of the $N_{CORE}$ domain or $P_{XD}$ itself).

## Discussion

To understand the role of disordered regions flanking the MoRE on either side in $N_{TAIL}$ dynamics and conformational preferences, we probed non-local interactions of the full-length protein in solution, combining PET experiments, analytical polymer models, MDs simulations, and coevolutionary analysis. With PET experiments, we measured and compared contact formation times between tryptophan and cysteine residue pairs introduced in different regions of the $N_{TAIL}$ sequence, under varying solution conditions. The experiments revealed a pronounced dynamic heterogeneity within the protein (Fig. 1) under physiological conditions, especially at the $N_{TAIL}$ C-terminal region, close to the MoRE, which is involved in binding to $P_{XD}$. PET experiments also showed a reduced heterogeneity under conditions that weaken electrostatic interactions, such as increased salt concentration or lower pH, pointing to an important role of electrostatic interactions between charged residues in the dynamics of $N_{TAIL}$. Based on our MDs simulations, we attribute this heterogeneity to salt bridge-induced non-local interactions between specific regions of $N_{TAIL}$.

### Interpretation of the general role of electrostatics

This finding is supported and explained in structural terms by our simulations (both coarse-grained and all-atom). Even without including atomic-level details, our CG simulations suggest that when increasing the electrostatic screening or changing the charge of acidic residues to emulate pH-related protonation, the N-terminal region expands while the C-terminal region compacts. In addition, CG simulations performed with different helical propensities in the MoRE indicate that changes in helicity of the MoRE do not give rise to the dynamic heterogeneity within the $N_{TAIL}$ C-terminal region (i.e., C3W2 and C4W2). These results indicate that side-chain specific transient interactions, not described by the CG model, contribute to the large difference between the C3W2 and C4W2 relaxation times.

### Specific non-local interactions dictate $N_{TAIL}$ conformation and dynamics

Our all-atom simulations, performed with a carefully validated force field, describe interactions with atomic-level detail. These simulations reproduce the dynamic heterogeneity of C-W pairs encompassing the $N_{TAIL}$ MoRE

(C3W2 compared to C4W2). We found that, while $N_{TAIL}$ is clearly disordered, a few key non-local interactions stabilize four distinct conformational states, which in total represent 48% of the structural ensemble. The two most populated states do not allow PET for either C3W2 or C4W2 variants, while the third and fourth states allow PET for the C4W2 and C3W2 variants, respectively. The free-energy landscape of $N_{TAIL}$ (Fig. 3) reveals rapid interconversions between the three most populated states. In contrast, structural rearrangements required to reach the C3W2 contact state are slower due to a small, though comparatively higher free-energy barrier (~4.4 kJ/mol) under physiological conditions. This free-energy barrier and the lower equilibrium population of the C3W2 contact state contribute to the dynamic heterogeneity of the $N_{TAIL}$ C-terminal region.

Analysis of the state-specific contacts (Fig. 4) reveals that competing clusters of interactions stabilize the four conformational states. These interaction clusters involve (and are possibly nucleated by) stable salt bridges between charged amino-acid side chains. Further, focusing on the two most populated conformational states S1 and S2, our all-atom simulations show that specific charged residues in region B (residues 435–451) can form either non-local salt bridges with C-terminal regions D-E (484–525, close to the MoRE), or local salt bridges with other region B residues. Switching between these sets of interactions (as sketched in Fig. 5) is necessary for $N_{TAIL}$ to transition between the two dominant states, affecting its large-scale dynamics.

Co-evolutionary analysis independently shows that correlated mutations across regions were only found between regions B and D. Given the distance in sequence between these two regions, this not only corroborates the presence of non-local interactions between region B and D, but also suggests that they play a functionally important role.

In summary, our simulations and co-evolutionary analysis independently highlighted region B of $N_{TAIL}$ as a potential allosteric interaction partner to region D, which includes the binding site (MoRE) for the phosphoprotein X domain ($P_{XD}$). Additionally, fluorescence-based binding kinetics experiments on truncated $N_{TAIL}$ variants[23], indicated that intramolecular interactions of $N_{TAIL}$ involving residues 435–451 (region B) have a significant impact on $P_{XD}$ binding in vitro, despite being located far away in sequence from the MoRE.

### Identified interactions in the context of $N_{TAIL}$ to $P_{XD}$ binding

We hypothesize that the intramolecular $N_{TAIL}$ interactions observed in our simulations not only govern the $N_{TAIL}$ dynamics but also play a role in regulating $P_{XD}$ binding to the MoRE. Mutational studies by Bignon et al.[57] have shown that the helix content of the MoRE correlates with the binding affinity, and that N-terminal truncation of $N_{TAIL}$ increases its affinity for $P_{XD}$[22,24,27]. Similarly, Gruet et al.[27] have identified mutations in several regions of the $N_{TAIL}$ sequence, on either side of the MoRE, which increase $P_{XD}$ binding affinity. The crystallographic structure of the $N_{TAIL}$ MoRE bound to $P_{XD}$[9] (PDB: 1t6o, Fig. 6B) shows that the binding interface consists mainly of a hydrophobic edge along the MoRE helix (region D), between residues S491 and M501 (hereafter D1 site, shown in dark green), which interacts via intermolecular contacts with the α2–α3 face of the $P_{XD}$ triple helix (Fig. 6A). The other side of the MoRE helix is a hydrophilic edge formed by charged and polar residues (D2 site, dark red) between Q486 and R497. The $N_{TAIL}$-$P_{XD}$ complex is further stabilized by intermolecular interactions involving R497 and salt bridges formed by D487 and R490, which are located at the beginning of the MoRE helix between the two edges (D3 site, light green)[27]. Many of these key amino acids are involved in intramolecular $N_{TAIL}$ interactions in one or more conformational states, as shown by the contact probabilities obtained from all-atom simulations (Fig. 4).

Based on these interactions, we propose two independent mechanisms that provide a possible explanation for the observed changes in $P_{XD}$ binding affinity upon $N_{TAIL}$ mutation or truncation.

### Direct competition for MoRE

In all-atom simulations, the D1 site residues (Fig. 6A, dark green) form intramolecular interactions with

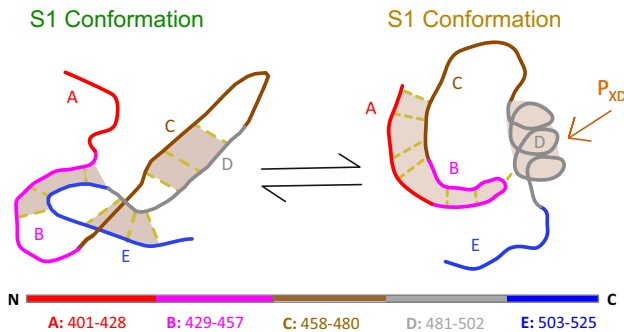

**Fig. 5 | Schematic representation of the two major $N_{TAIL}$ conformational states (S1 and S2).** Different regions of $N_{TAIL}$ are color-coded according to Fig. 4. Non-local interactions between the regions are represented as dashed lines. Key interactions between regions are highlighted by orange shading. The helical binding site of the phosphoprotein X domain ($P_{XD}$) in region D is marked by an orange arrow.

residues of regions B and C in state S2 as well as with residues of region E in state S3 (see Supporting Table S7), both of which are states with considerable helical propensity of the MoRE region. Assuming that a folded MoRE helix is a necessary part of the biologically active $N_{TAIL}$ conformation, these intramolecular interactions would directly compete with the intermolecular interactions with $P_{XD}$.

First, the intramolecular interactions would reduce the free helix population of the MoRE, thereby decreasing the binding rate of $P_{XD}$. Second, the intramolecular interactions may also displace $P_{XD}$ from the binding site without exposing the hydrophobic D1 site to water, thereby lowering the $P_{XD}$ unbinding activation barrier and increasing the unbinding rate. Both effects would regulate MoRE-$P_{XD}$ binding by reducing the binding affinity, and explain why the binding affinity increases when flanking regions of the MoRE are removed[27,57].

### Indirect allosteric regulation

Helical conformations of the $N_{TAIL}$ MoRE appear to have a higher affinity for $P_{XD}$[57]. Therefore, changing the population of helical MoRE conformations via intramolecular interactions would also affect the MoRE-$P_{XD}$ binding affinity. This mechanism is similar to the ensemble allosteric model for IDPs described by White et al.[29]. Specifically, intramolecular interactions that stabilize non-helical MoRE conformations, such as the contacts seen in state S1, would decrease the population of helical conformations, reducing both $N_{TAIL}$ binding affinity and the rate of binding to $P_{XD}$. The observed contacts (Fig. 4) suggest that the ratio of states S1 and S2 is controlled by the competing interactions of region B, which acts as a conformational switch. Specifically, in state S1, the interactions between regions B and E stabilize a loop-like, non-helical conformation of the $N_{TAIL}$ C-terminus, which is likely binding-incompetent toward $P_{XD}$. In contrast, in state S2, region B residues engage in local interactions, allowing the $N_{TAIL}$ C-terminus to form the high-affinity helical structure in the MoRE at region D.

### Potential functional role of $N_{TAIL}$ intramolecular interactions in viral replication

Non-local interactions between regions B and D/E of $N_{TAIL}$ may also play a functional role in RNA polymerization and viral replication. In the following, we present three established mechanisms where intramolecular interactions between the B and D regions of NTAIL may contribute to optimising the binding kinetics for viral replication.

### Tether shortening

Initial polymerase complex recruitment likely involves the formation of one or more disordered tethers between the polymerase complex and the nucleocapsid. By limiting the volume available for diffusion, tethering increases the local concentration of the polymerase complex around the nucleocapsid, thereby increasing

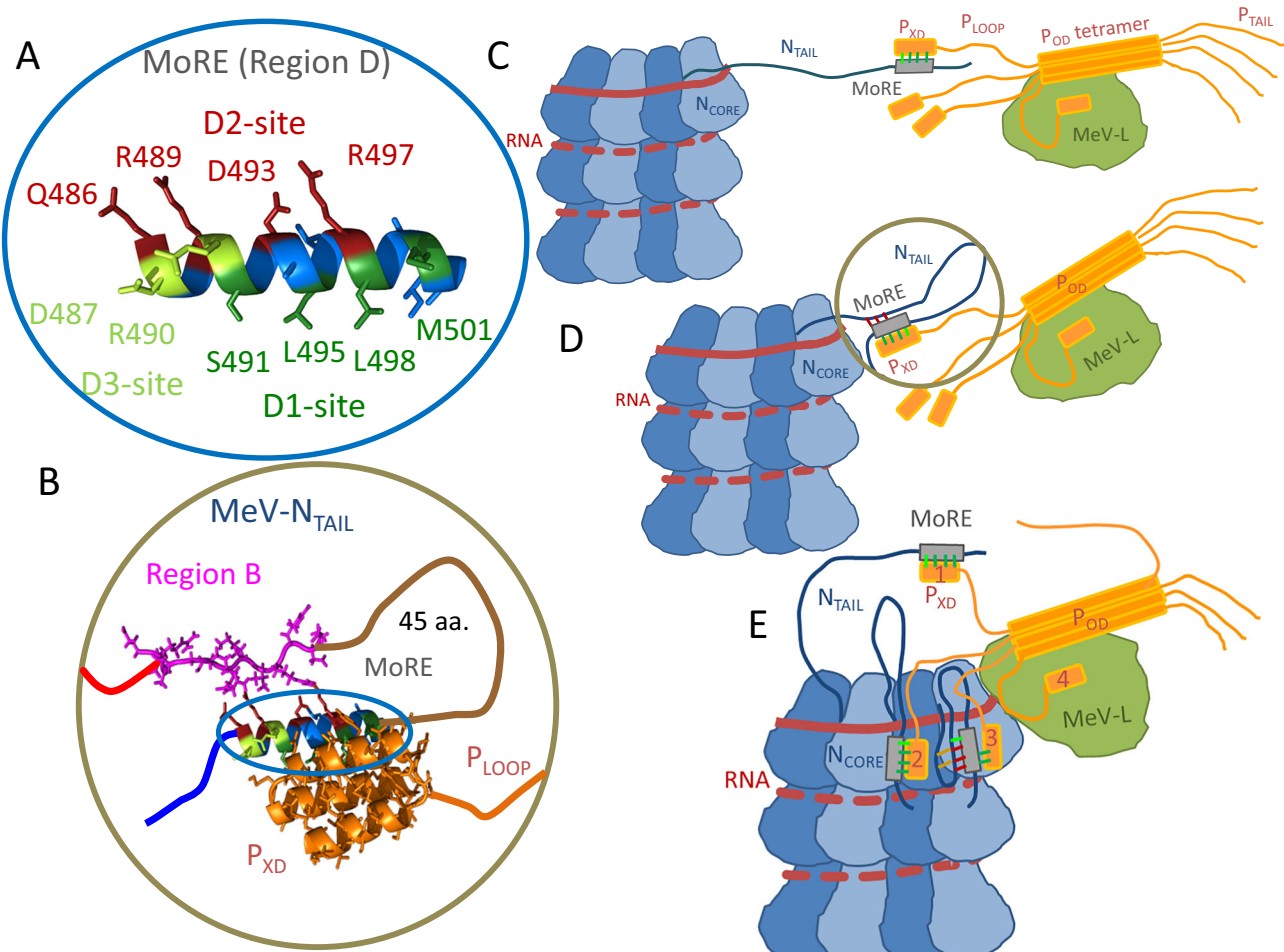

**Fig. 6 | Speculated biological context of non-local $N_{TAIL}$ interactions. A** $N_{TAIL}$ MoRE residues in region D constituting the hydrophobic $P_{XD}$-binding interface are highlighted (D1 site, dark green) along with charged residues possibly involved in intramolecular $N_{TAIL}$ interactions (D2 site, dark red), and residues possibly involved in both intra- and intermolecular interactions (D3 site, light green). **B** Cartoon representation of the $P_{XD}$-MoRE complex based on the available crystal structure (PDB: 1t6o). **C** During the initial steps of the virus replication the MeV nucleocapsid protein (N, blue) recruits the large polymerase (L, green), by binding to the phosphoprotein (P, orange). Here, $N_{TAIL}$ is extended. **D** When Brownian motion allows L to approach the RNA (red) in the nucleocapsid groove, $N_{TAIL}$ adopts a hairpin conformation, which promotes intramolecular interactions between the MoRE and the N-terminal region of $N_{TAIL}$. **E** Once the contact between nucleocapsid and L is established, additional complexes can be formed between free $P_{XD}$ and downstream N monomers. Further, intramolecular $N_{TAIL}$ interactions help release bound P to allow polymerase progression by weakening the binding to $P_{XD}$ when L is nearby. Numbers (1–4) on $P_{XD}$ denote which step individual domains are in a "cartwheeling" L progression mechanism. **C–E** Only 1–3 $N_{TAIL}$ domains have been depicted for the sake of clarity. The RNA molecule is shown as a solid line on its first helix turn around the nucleocapsid to indicate that it is more solvent exposed, and as a dashed line on downstream turns to highlight that it is partially buried in the RNA-binding ridge.

the rate of transcription/replication initiation. There are two experimentally established tethers between N and P. The first tether consists of the disordered N-terminal tail of P (1–304, $P_{TAIL}$) binding to two opposite sides of $N_{CORE}$, and its primary function appears to be the prevention of nucleocapsid association in the absence of the viral RNA[58]. The second tether is formed by regions B and C of $N_{TAIL}$ and the C-terminal domain ($P_{LOOP}$, 376-459) of P, through coupled binding and folding between the MoRE of N and $P_{XD}$ (Fig. 6C). This tether regulates L progression on the nucleocapsid[59]. The two tethers may coexist and, in fact, work in concert to form N-P condensates that serve as the basis of viral factories within the host cell[60,61], and to securely attach the polymerase complex to the nucleocapsid during transcription/replication initiation[11].

During replication initiation, it is preferable to have long tethers to recruit polymerase complexes from a larger volume around the nucleocapsid, but later, when L has to be guided to the 3' end of the RNA, shorter tether lengths may be preferable. Intramolecular $N_{TAIL}$ interactions between the MoRE D2 site and region B would shorten the tether by 45 amino acids

(Fig. 6D), further reducing the volume available to the polymerase complex, and increasing the rate of initiation of genome transcription and/or replication.

**$P_{XD}$ unbinding during polymerase progression.** Sourimant et al.[10] carried out recombinant minigenome bioactivity assays and found that polymerase progression is slowed down considerably by a deletion of regions B and C of $N_{TAIL}$, and proposed a mechanism for polymerase progression. This mechanism is based on a newly discovered binding site on $N_{CORE}$ for the α1-α2 face of $P_{XD}$, at which compensatory point mutations could weaken $P_{XD}$-$N_{CORE}$ binding and restore bioactivity. Briefly, the mechanism suggests that the L-P polymerase complex is tethered to the nucleocapsid by forming the $N_{TAIL}$ MoRE—$P_{XD}$ complex. Then, the $P_{XD}$-MoRE complex binds the recently discovered $N_{CORE}$ site. $P_{XD}$ binding to this $N_{CORE}$ site likely facilitates $P_{XD}$-MoRE unbinding, possibly after compaction of the $N_{TAIL}$ loop to allow polymerase passage. $P_{XD}$ unbinding in turn allows subsequent binding to MoRE of the downstream N monomer and polymerase translocation.

The mechanism proposed by Sourimant et al. provides previously unprecedented molecular insight into the MeV replication. However, it mostly considers the deleted regions of $N_{TAIL}$ as passive linkers or "roadblocks" for L, and it does not give an in-depth explanation on how weakening the $N_{CORE}$-$P_{XD}$ binding affinity compensates for reduced bioactivity caused by the deleted $N_{TAIL}$ segments. Our current results and previous binding affinity studies on truncated $N_{TAIL}$ constructs[22] suggest that non-local interactions between MoRE and $N_{TAIL}$ regions B and C can, in fact, actively compete with $P_{XD}$ for its binding site and may do so in the context of a MoRE-$P_{XD}$-$N_{CORE}$ ternary complex as well. This implies that deleting regions B and C does not just shorten $N_{TAIL}$ but also eliminates intramolecular interactions that promote MoRE-$P_{XD}$ unbinding. This would explain why the deletion hampers viral replication, why mutations at the $N_{CORE}$ interface compensate for the deletion, and provides a possible reason for region B residues to co-evolve with the MoRE (Table S5). In addition, based on the available cryo-EM structure of the nucleocapsid[1] (PDB: 4uft), region B would emerge from the RNA-binding ridge close to the $N_{CORE}$ binding site, with sufficient space for region B to fold into a loop and interact with the MoRE region (Fig. S24).

**Polymerase progression by cartwheeling.** Our findings are also compatible with a modified version of the cartwheeling mechanism[62] for polymerase progression (Fig. 6E). Here, P is recruited (Fig. 6E, step 1) by $N_{TAIL}$ and bound to the $N_{CORE}$ binding site (step 2) of the same monomer as described by Sourimant et al.[10]. However, in this mechanism, $P_{XD}$ remains bound to the same N monomer until the L polymerase approaches. Then, due to steric hindrance from L or conformational changes at the N-N interface, $N_{TAIL}$ folds back onto the nucleocapsid (step 3) in a conformation that allows region B to weaken $P_{XD}$-MoRE interactions. This, in turn, allows $P_{XD}$ to unbind and be transferred to its binding site on L (step 4) as L progresses. Subsequently, $P_{XD}$ is either released to the solvent or directly binds to the $N_{TAIL}$ of a downstream N monomer in the next cartwheeling cycle.

As discussed above, evidence suggests intramolecular $N_{TAIL}$ interactions weaken $P_{XD}$ binding. Tight binding to $P_{XD}$ is beneficial for tethering the viral polymerase to the nucleocapsid during polymerase recruitment and progression, and thereby ensures that the transcription and replication of the genome are completed. However, unbinding of individual $P_{XD}$-$N_{TAIL}$ complexes is required for polymerase progression during viral replication[63]. Therefore, MeV and related viruses may be at an evolutionary disadvantage if the binding between $P_{XD}$ and $N_{TAIL}$ is so tight that it limits the replication rate, as supported by mutational studies of $P_{XD}$[64]. It is thus tempting to hypothesize that a possible functional role of non-local intra-protein $N_{TAIL}$ interactions is to provide a conformation-dependent optimization of $P_{XD}$ binding affinity.

## Conclusions

Our combined experimental and simulation approach shows that $N_{TAIL}$ is clearly disordered, contains very little secondary structure, and shows nearly diffusive dynamics. However, it still displays some structural features which cause its dynamics to deviate significantly from those of a homopolymer. In particular, we identified several key transient interactions between disordered regions distant in sequence as the main reason for this deviation. These interactions affect the overall conformation and dynamics of $N_{TAIL}$ in solution. Interestingly, similar interactions also emerge from our independent co-evolutionary analysis, corroborating our simulation results and suggesting that they not only govern the conformational dynamics of the essential $N_{TAIL}$ domain, but also are functionally important. We therefore propose possible mechanisms by which these non-local interactions regulate binding to the $P_{XD}$, and consequently, recruitment and progression of the polymerase complex onto the nucleocapsid template. Our results for $N_{TAIL}$ corroborate the importance of flanking regions in IDPs that bind via short molecular recognition elements (MoREs)[64], and suggest a regulatory role of regions that may be far in the sequence from MoREs. These non-local interactions within $N_{TAIL}$ can regulate binding via an ensemble allosteric

model[29,30] without requiring the binding of a third molecule. Further studies targeting the $N_{TAIL}$-$P_{XD}$ interaction may provide additional insight into the underlying mechanisms and effects of these interactions.

It is plausible that similar mechanisms are also at work for other IDPs that share sequence characteristics, such as charge patterning, with $N_{TAIL}$. Specifically, non-local intra-protein interactions in IDPs may regulate the conformational preferences and dynamics of both the MoRE and of the entire protein, either via direct competition or indirectly by shifting the population of free-energy minima.

## Methods

### Protein expression, purification, and sample preparation

Wild type (WT) $N_{TAIL}$ and the seven variants listed in Supporting Table S1, were expressed in *E. coli*, and further purified as described in detail in Supporting Methods S1. Briefly, after expression of tagged protein in *E. coli*, cell pellets were resuspended in 8 M urea, 50 mM Tris pH8, 0.3 M NaCl, and frozen. After thawing, the solution was sonicated, spun, and tagged proteins were purified using Ni-Sepharose fast flow beads (Cytiva). After dialyzing the eluent in 50 mM Tris pH8, 0.3 M NaCl, TEV protease was used to remove tags from the desired protein. TEV protease, as well as uncut tagged proteins, were eventually removed using Ni-sepharose beads. Purified $N_{TAIL}$ proteins, free of histidine tags, were dialyzed against 15 mM Tris, 150 mM NaCl, 1 mM TCEP buffer at neutral pH, checked using SDS-PAGE, UV absorption, and far-UV CD, and frozen for shipping. After shipping, samples were further purified via HPLC (semipreparative Vydac C18 column) and lyophilized. Before PET experiments, the lyophilized protein was dissolved directly into filtered buffers (for reference conditions: 15 mM Tris, 150 mM NaCl, 1 mM TCEP pH 7.6; for low pH measurements: 20 mM NaAc, 150 mM NaCl, 1 mM TCEP pH 4.0; for high salt concentrations: 15 mM Tris, 500 mM NaCl, 1 mM TCEP pH 7.6). Protein concentration was adjusted to ~100 μM as determined by UV absorbance at 280 nm (extinction coefficient 6990 cm$^{-1}$ M$^{-1}$). Samples (~350 μL) were placed in 5 × 10 × 30(h) mm Spectrocell quartz gas tight cuvettes with screwcap (equipped with Teflon coated silicon membrane), and bubbled with USP grade nitrous oxide for at least one hour, to reduce the concentration of dissolved oxygen (a quencher of excited tryptophan triplet state), and to introduce solvated electron scavenger (to prevent reactions with electrons, produced by water decomposition under UV laser pulses).

### PET experiments

To probe intramolecular contact formation, we used a technique based on PET between a tryptophan (W) and a cysteine (C)[34–36] placed at different positions within the sequence (sequences in Supporting Table S1). Details are described in Supporting Methods S2. Briefly, we used a homebuilt nanosecond transient absorption apparatus to excite the W to the triplet state and to monitor the excited state population as a function of time. When C comes into contact with W (within the Van der Waals distance) due to stochastic collisions, an excited-state electron is transferred from the triplet state of W to C. As explained in Supporting Methods S2, and in Sizemore et al.[36], the measured relaxation time is related to the intra-molecular contact formation time between W and C in the protein. The contribution to the triplet state relaxation times due to C-W quenching via electron transfer, are reported in Fig. 1D. These were obtained by fitting W triplet state relaxation curves as illustrated in Supporting Figs. S1 and S2, and taking into account the natural lifetime of the triplet state in the absence of C, measured on single W variants (see Supporting Methods S2). Each value is the result of globally fitting multiple repeated measurements under given solution conditions. All raw data and their fits are shown in Supporting Figs. S3 through S16. The corresponding fitting parameters, obtained for each variant under each solution condition, are reported in Supporting Table S2. Solution conditions used in PET experiments of Fig. 1 were, for reference conditions: 15 mM Tris, 150 mM NaCl, 1 mM TCEP pH 7.6; for low pH measurements: 20 mM NaAc, 150 mM NaCl, 1 mM TCEP pH 4.0; for high salt concentrations: 15 mM Tris, 500 mM NaCl, 1 mM TCEP pH 7.6.

## Circular dichroism

CD spectra reported in Fig. 1D inset were measured as described in Supporting Methods S3. To improve signal-to-noise and access low wavelengths, which are particularly important for the interpretation of IDP spectra, at each pH, we carried out measurements for a series of protein concentrations and combined them. Measurements at high salt concentration were not possible due to the high absorbance of these samples below 210 nm, a range which is most important for the interpretation of IDP spectra. Solution conditions used for spectra of Fig. 1 were: 10 mM $NaPO_4$, 150 mM, NaF for pH 7.6 measurements, and 20 mM NaAc, 150 mM NaF for pH 4.0 measurements.

## Coarse-grained MDs

We applied a coarse-grained model based on the original HPS model[43] to $N_{TAIL}$. Each amino acid is represented by a bead with charge $(+1, 0, -1)$ and hydropathy. There are three types of interactions in the HPS model: bonded interactions, electrostatic interactions, and short-range pairwise interactions. The electrostatic interactions are modeled using a Coulombic term with Debye-Hückel[43] electrostatic screening to account for salt concentration. The short-range pairwise potential accounts for both protein-protein and protein-solvent interactions, which was optimized using the experimental radius of gyration of $N_{TAIL}$[6]. We further added additional terms for angle and dihedral preference so that the secondary structure of the MoRE region can be captured (see Supporting Methods S5). The HOOMD-Blue software v2.9.3[65] together with the azplugins (https://github.com/mphowardlab/azplugins) was used for running the MDs simulations. All simulations were run using a Langevin thermostat with a friction coefficient of 0.01 $ps^{-1}$, a time step of 10 fs and a temperature of 298 K for 5 μs . The first 0.5 μs were discarded for equilibration.

## All-atom MDs

We performed all MD simulations on three full-length $N_{TAIL}$ variants. The wild-type (WT) NTAIL (401–525) simulations were performed, including an N-terminal hexa-histidine (His6) tag, while the latter was not included in the case of variants C3W2 (i.e., S482C-Y518W) and C4W2 (i.e., S488C-Y518W) to match the available experimental data as much as possible. All simulations were performed using the GROMACS 2019 simulation package[66]. Each $N_{TAIL}$ variant was simulated using two different force fields (CHARMM36m[51] and AMBER99SB-disp[49]), and at least three replicas per force field started from different initial conformations. The WT $N_{TAIL}$ simulations were performed only with the CHARMM36m force field. For the Amber force field validation, we used previously published wild-type $N_{TAIL}$ simulation trajectories[26]. The simulations of the two variants in Amber were carried out using force field parameters adapted to the GROMACS simulation package. Both the WT trajectories and the force field parameters were kindly provided by Piana et al. Each simulation was carried out using periodic boundary conditions in a box filled with explicit-solvent molecules consisting of either OPC[55] (for CHARMM36m) or a modified four point-charge (TIP4P[50] for AMBER99SB-disp) water models as well as Na+ and Cl− ions corresponding to an ion concentration of 150 mM. The total size of simulated systems was ~120,000 to 200,000 atoms. Total simulation time was >40 μs per variant, with conformations recorded after every nanosecond. Hydrogen bond vibrations were constrained using virtual atom sites to enable a 4 fs time step during simulations. All simulations were kept at 1 atm pressure and 298 K temperature. Further details on the MD simulation parameters and system preparation are provided in the Supporting Methods S7 and S9. Analysis methods related to the comparison to experimental data, calculation of mutual information, contact maps, and free-energy landscapes are described in the Supporting Methods S11, S19-S21.

## Reporting summary

Further information on research design is available in the Nature Portfolio Reporting Summary linked to this article.

## Data availability

The implementation of the coarse-grained HPS model can be downloaded from https://github.com/wzhenglab/ntail. Other data supporting the findings of this study are available from the corresponding authors upon reasonable request.

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

## Acknowledgements

L.O., J.K., G.K., S.L. and S.M.V. acknowledge the support from the National Institutes of Health (R01GM120537). A.C.V., H.G., L.V.B. and G.N. were supported by the Max Planck Society and by the German Science Foundation (DFG), Excellence Strategy Grant MBExC 2067/1; computer time was provided by the Max Planck Computing and Data Facility. G.N. was additionally supported by the Alexander von Humboldt Foundation. A.C.V. was additionally supported by the European Union's Horizon 2020 Framework Program for Research and Innovation, Specific Grant Agreement No. 945539 (Human Brain Project SGA3). W.Z. acknowledges the support from the National Institutes of Health (R35GM146814) and the research computing at Arizona State University.

## Author contributions

L.O. and G.K. carried out, analyzed and interpreted PET and CD experiments and contributed to writing the manuscript. J.K. carried out initial PET and CD experiments. S.M.V. conceived the original design, supervised the PET and CD experiments, contributed to conceptualization and interpretation, and wrote the manuscript. C.B. generated the expression constructs, protein expression and purification. S.L. contributed to the original design of expression constructs, supervised the protein expression and purification, and contributed to editing the manuscript. W.Z. performed and analyzed the coarse-grained simulations, and co-evolution analysis and contributed to conceptualization, interpretation and writing the manuscript. G.N. carried out, analyzed, and interpreted the atomistic simulations, analyzed CD spectra, and contributed to the conceptualization and writing the manuscript. L.V.B. contributed to the interpretation of the simulation data and to writing the manuscript. A.C.V. and H.G. contributed to supervision of the atomistic simulations, conceptualization and interpretation, and to writing the manuscript.

## Funding

## Competing interests

The authors declare no competing interests.
