## [Transparent Peer Review file · Communications Chemistry]

Transient Non-local Interactions Dominate the Dynamics of Measles Virus N_{TAIL}

Corresponding Author: Dr Sara Vaiana

Version 0:

Reviewer comments:

Reviewer #1

(Remarks to the Author)

This paper is a well-executed exploration of how transient non-local interactions shape the behavior of the measles virus N_{TAIL} domain. The authors do an excellent job combining experimental techniques with computational modeling to uncover previously unknown interactions that influence the structure and function of this intrinsically disordered protein. Their findings not only deepen our understanding of N_{TAIL}'s structural properties but also shed light on its crucial role in viral transcription and replication.

The paper is surely well written, the conclusions are sound and well explained. The use of photo-induced electron transfer (PET) experiments allows for an unusually precise look at how different regions of N_{TAIL} interact in real time—an approach that overcomes many of the challenges associated with studying intrinsically disordered proteins. The authors go a step further by validating their experimental results with a combination of coarse-grained and all-atom molecular dynamics simulations, carefully ensuring that their computational models align with real-world data. Their attention to force field validation and systematic cross-checking with experimental results makes this an exceptionally robust and convincing study.

Beyond just the methods, the biological significance of the findings is also appealing. The discovery that N_{TAIL}'s conformational behavior is shaped by non-local electrostatic interactions provides a new perspective on how this domain regulates binding to the phosphoprotein X-domain (PXD) and, ultimately, the recruitment of the viral polymerase. The way in which specific charged residues influence the structural preferences of N_{TAIL}—and how this is further supported by coevolutionary analysis—adds another layer of importance to the work.

I can only recommend acceptance

Reviewer #2

(Remarks to the Author)

In this work, the authors present a compelling and impactful set of experiments and simulations to determine the role of flanking regions in the regulation of N_{tail}. They conclude that transient interactions driven by electrostatics in the flanking regions have a significant effect on N_{tail} conformations and thus function. The PET experiments show clear heteropolymeric behavior, with charge interactions in two regions of the protein having opposing effect. Both the coarse grained and all atom simulations bring complementary information to understand the protein dynamics. In general, I found the overall work and conclusions to be very robust. I had only fairly minor suggestions or questions.

There appear to be several steps in converting between simulation and PET experiments, and so plotting the ratio of the different rates is important. In Figure 2B, the ratio is between the reference condition and different buffer conditions was used, however a more relevant comparison might be the differences between PET constructs. Without this, it's hard to determine how well the simulations reproduce either the overall trend or the deviations relative to the trend as in figure 1C. Can you plot something like Figure 1C for both types of simulations? Even if it's somewhat different, e.g. the appropriate distance distribution between relative amino acids even if they aren't mutated in the simulations? That is, comparisons like Figure S22, but for all of the constructs seem very important for validating the comparison between experiment and simulations.

The analysis on the four conformations in the all atom simulations is quite nice, but it would also be nice to see a similar analysis averaged across the whole simulation. Contact maps or analysis like figure 2A would both be very useful.

It would be nice to include a small figure of charge density along the sequence, that is the diagonal of Fig 2 top is important enough to take a bit more space.

I don't understand the charge state of D and E residues at pH 4 in the coarse grained simulation. Are they chosen randomly to be charged or not charged and then fixed at that charge, given fractional charges, or stochastically changed throughout the simulation. The above plot could include the simulation charge distribution both at physiological pH and at pH 4.

It would be nice if figure 2C could include comparisons to experiment for the salt condition as well, although I'm unclear if that experiment has been done. Perhaps it is a different ratio. The presentation is actually a bit confusing, the dots in figure 2B are hard to see, but clearer than the lines in Figure 2C, since the 2C lines are only comparisons to a single condition.

The calculated bare rates for quenching as a function of pH are the same within experimental error, but still almost 20% different. Is it possible to get more accuracy on these measurements? It might be useful to correct the comparisons between the protein PET results.

The all atom simulations were (understandably) only done at one pH, and if I follow the logic correctly, the argument is that since the "physiological" PET rates being so very different between C3W2 and C4W2 is recapitulated in the all atom model, then they are relevant. However, the coarse-grained model were also able to reproduce that difference between rates, but unable to capture the changes due to the pH shift. This is all fine, but from a writing perspective, it makes it unclear whether the all atom have really succeeded in reproducing the experimental data. I wonder, simply from a story-telling perspective, if it would better to focus on the increased sequence information given by the all atom rather than the limitations of the coarse grained model.

Reviewer #3

(Remarks to the Author)

In this study, the authors explored how certain internal interactions within the measles virus nucleoprotein (NTAIL) affect its behavior and role, especially in its connection with the polymerase complex, which helps the virus replicate and transcribe. They used experimental and computational methods and discovered that short-lived interactions between separate parts of NTAIL are crucial in shaping its behavior and interactions with other proteins, influencing the virus's ability to replicate and transcribe. The findings suggest these types of interactions might also be important in other similar proteins, although this was not directly investigated or substantiated.

Overall the paper provides a thorough and valuable analysis that contributes significantly to our understanding of these proteins and should definitely be published.

Concerns:

While I have no major concerns about the scientific value or quality of the research, the presentation could be clearer and less technical. I honestly had a hard time following many parts of the text and managing the large number of supplementary figures and tables. I know they are important, but they sometimes made it hard to follow the logic of the paper. In summary, many times I felt that the paper was too technical for a broader chemistry audience journal.

On the other hand, I missed some information that would be better presented in a table. For greater clarity on the computational work, it would help to include details like the duration of each simulation in a table. This table could also specify the number of replicates and other details that distinguish each simulation. This would make it easier for others to replicate the study and to assess whether the results are based on a single analysis or a robust study with multiple replicates that also considers the statistical significance of the results. I honestly don't know at this point the answer for this last point.

A couple of specific minor suggestions:

1. Rearrange Figure S24 to improve visibility. Consider placing part A above part B and enlarging A, as it is quite hard to see the details described on page 35.
2. In Table S1, please clarify what the colors represent.

Reviewer #4

(Remarks to the Author)

The manuscript by Otteson et al. the interactions between amino acid residues that are distant in sequence of the N_tail protein of the measles virus are investigated using photo-induced electron transfer (PET) and Molecular Dynamics (MD) simulations.

From detected PETs and analysis of extensive MD simulations, the temporary and short-lived interactions between introduced tryptophan and cysteine pairs can be seen. An estimate of the rate of conversion between states S1-S4 from a Markov model would be beneficial but is not absolutely critical here.

This is interesting since the intrinsically disordered protein (IDP) N_tail is so flexible and showing a multi-dimensional conformational surface, that this information cannot be obtained otherwise.

It is clear that the authors extend the system beyond the MoRE region flanked by residues of N_tail to either side but we would need to see a discussion of previous work by NMR and all-atom MD (their references 19 and 26). Does this region retain its secondary structure in solution in experiment and solution and how do the results of this work agree with refs 19 and 26.

In experiment and MD simulations, the authors show that it is electrostatic interactions between differently positioned amino acid residues, which is responsible for the heteropolymeric behavior of N_tail.

It remains to be mentioned critically, that the first coarse-grain (MD) simulations required the adjustment of parameters to be able to get the radius of gyration and the helical propensity of the MoRE region close to experimental values from SAXS and NMR.

When altering the pH in experiment and simulations, a lower pH of 4.0 leads to 'more neutral' Asp and Glu residues and neutral His will become 'more positively charged'. This statement is correct but a quantification is needed, how many protons will be added to these residues at pH 4.0. It requires an estimation of the pKa values of the individual residues and then an annotation of change of N_tail residue by pKa.

On the other hand, on p 13, line 9 the authors state that changes in PET are 'entirely due to changes in the protein, rather than pH-induced changes of the electron transfer efficiency'. It is clear that the first will effect the latter. pH-induced changes in protein structure will critically effect the distances between W and C residues.

In Figure 4, the side chain salt bridge and C_alpha backbone interactions are shown as contact maps. These are very hard to see and a better coloring scheme or radius is required.

On p 18, line 3, the authors claim to have validated the choice of force field for the protein and water against experimental CD and PET experiments. However, we see only two combinations CHARMM36m/OPC and AMBER99SB-disp/TIP4P in the methods part. The authors need to clarify this.

In Figure 5, the dashed lines are not visible and the figure has to be improved in clarity.

The conclusion section spans from pages 26 to 36 and appears before the summary. First, the summary ought to be communicated to the reader and then a SHORT section of conclusions can be drawn to speculate about the results of this work on isolated N_tail with relevance to its physiological process. However, the latter is tentative only and should be presented in a concise way.

Overall, the study is interesting and demonstrates the complexity of the conformational dynamics and flexibility of intrinsically disordered proteins and their possible function in biological processes. Rationalizing PET experiments for two pairs of mutated C/W residues requires VERY extensive sampling by classical MD simulations.

Version 1:

Reviewer comments:

Reviewer #2

(Remarks to the Author)

The revisions significantly improved the manuscript and I have no further concerns.

Reviewer #3

(Remarks to the Author)

The revised form of the manuscript is much easier to read. Although the topic is quite complex, I believe it is now

significantly more accessible to a broad readership. I had no major concerns with the previous version, and I believe that the new revision only strengthens the presentation of the results. I recommend publication.

Reviewer #4

(Remarks to the Author)

The revised manuscript reads much better and all issues with the original were resolved. I can thus only recommend its publication.

Response to Referees:

We would like to thank the Reviewers for their hard work and constructive comments. In response to the Reviewer's comments, we extensively modified our manuscript, along three main lines:

- we clarified the text of the manuscript to make it more accessible to a broader audience, significantly rewriting entire sections, and reorganizing the material to simplify the reading,
- we added short summaries of our methodology in the main text, wherever needed to allow the reader to understand the essentials of our results without having to rely on the supplementary materials,
- we improved the text to avoid technical jargon.

In the revised manuscript, all changes to the text are highlighted in red.

In addition to these three main issues, we addressed all specific comments, of each reviewer as described in detail below.

Reviewer #1

This paper is a well-executed exploration of how transient non-local interactions shape the behavior of the measles virus NTAIL domain. The authors do an excellent job combining experimental techniques with computational modeling to uncover previously unknown interactions that influence the structure and function of this intrinsically disordered protein. Their findings not only deepen our understanding of NTAIL's structural properties but also shed light on its crucial role in viral transcription and replication.

The paper is surely well written, the conclusions are sound and well explained. The use of photo-induced electron transfer (PET) experiments allows for an unusually precise look at how different regions of NTAIL interact in real time—an approach that overcomes many of the challenges associated with studying intrinsically disordered proteins. The authors go a step further by validating their experimental results with a combination of coarse-grained and all-atom molecular dynamics simulations, carefully ensuring that their computational models align with real-world data. Their attention to force field validation and systematic cross-checking with experimental results makes this an exceptionally robust and convincing study.

Beyond just the methods, the biological significance of the findings is also appealing. The discovery that NTAIL's conformational behavior is shaped by non-local electrostatic interactions provides a new perspective on how this domain regulates binding to the phosphoprotein X-domain (PXD) and, ultimately, the recruitment of the viral polymerase. The way in which specific charged residues influence the structural preferences of NTAIL—and how this is further supported by coevolutionary analysis—adds another layer of importance to the work.

I can only recommend acceptance

We thank the reviewer for their detailed and positive evaluation of our manuscript!

Reviewer #2

In this work, the authors present a compelling and impactful set of experiments and simulations to determine the role of flanking regions in the regulation of Ntail. They conclude that transient interactions driven by electrostatics in the flanking regions have a significant effect on Ntail conformations and thus function. The PET experiments show clear heteropolymeric behavior, with charge interactions in two regions of the protein having opposing effect. Both the coarse grained and all atom simulations bring complementary information to understand the protein dynamics. In general, I found the overall work and conclusions to be very robust. I had only fairly minor suggestions or questions.

Comment 2-1:

There appear to be several steps in converting between simulation and PET experiments, and so plotting the ratio of the different rates is important. In Figure 2B, the ratio is between the reference condition and different buffer conditions was used, however a more relevant comparison might be the differences between PET constructs. Without this, it's hard to determine how well the simulations reproduce either the overall trend or the deviations relative to the trend as in figure 1C. Can you plot something like Figure 1C for both types of simulations? Even if it's somewhat different, e.g. the appropriate distance distribution between relative amino acids even if they aren't mutated in the simulations? That is, comparisons like Figure S22, but for all of the constructs seem very important for validating the comparison between experiment and simulations.

We appreciate the reviewer's insightful suggestion. Indeed, comparing across the five PET constructs, as done for the experiments in Figure 1C, would provide important context for validating the simulation trends. However, to determine the overall relaxation-time trends between PET constructs both of CG and all-atom simulations requires additional assumptions we were originally reluctant to make. Encouraged by the reviewer's suggestion, we have now added the requested comparisons, as well as the underlying assumptions.

For the CG simulations, computing absolute PET rates would require knowledge of prefactors, which are now discussed on page 15 of the revised manuscript. Although these prefactors cannot be determined from the simulations, in the revised manuscript, we have assumed that they are identical for all C-W pairs, and have estimated them by fitting a homopolymer model to CG simulation data, rescaling them as discussed in the revised manuscript. The resulting plot is now shown in Fig 2D.

For the all-atom simulations, we computed the absolute relaxation rates for C3W2 and C4W2, which are now added and discussed on page 18 of the revised manuscript. However, in order to compute relaxation rates for the other three variants (C1W1, C2W1, C3W1), we have to assume that the short-range contact dynamics is unaffected by Y->W mutations at the W1 position. Because the size and shape of the two side chains differ, our predicted PET relaxation times are probably less accurate even if the overall dynamics of Ntail is captured accurately by the simulations. To take this effect partially into account, we have now calculated the relaxation times for these variants using a slightly larger (0.6 nm instead of 0.4 nm) contact distance cutoff, as shown in the new supplementary Figure S30. The figure shows that the general trends are indeed captured by the all-atom simulations, although the C3W1 relaxation times are slightly overestimated, potentially due to insufficient sampling of the nonlocal interactions between the regions B and D.

Together, these figures allow for a more direct comparison to the experimental trends in Figure 1C (Fig. 1D in the revised version), and help assess to what extent the simulations capture both the expected scaling and the deviations across constructs. We believe this addresses the reviewer's

concern and strengthens the interpretation of our comparison between coarse-grained simulations and experiments.

Comment 2-2:

The analysis on the four conformations in the all atom simulations is quite nice, but it would also be nice to see a similar analysis averaged across the whole simulation. Contact maps or analysis like figure 2A would both be very useful.

We agree with the reviewer's suggestion, and now provide an averaged contact map in the supplementary information (Fig. S27).

Comment 2-3:

It would be nice to include a small figure of charge density along the sequence, that is the diagonal of Fig 2 top is important enough to take a bit more space.

We gladly agree, and have now added a net charge per residue plot in Fig. 1B.

Comment 2-4:

I don't understand the charge state of D and E residues at pH 4 in the coarse grained simulation. Are they chosen randomly to be charged or not charged and then fixed at that charge, given fractional charges, or stochastically changed throughout the simulation. The above plot could include the simulation charge distribution both at physiological pH and at pH 4.

The description of the charge treatment of ionizable residues in the coarse-grained simulations was not sufficiently clear. In our model, we assigned fixed fractional charges to titratable residues based on the Henderson-Hasselbalch equation $\text{pH} = \text{pK}_a + \log([\text{A}^-]/[\text{HA}])$ and pK_a values for side chains of Aspartic acid (3.65), Glutamic acid (4.25) and Histidine (6.00). We have now clarified this aspect on page 12 of the revised manuscript as well as in Supporting Methods S5.

Comment 2-5:

It would be nice if figure 2C could include comparisons to experiment for the salt condition as well, although I'm unclear if that experiment has been done. Perhaps it is a different ratio. The presentation is actually a bit confusing, the dots in figure 2B are hard to see, but clearer than the lines in Figure 2C, since the 2C lines are only comparisons to a single condition.

We agree with the reviewer that Fig. 2B and 2C were difficult to interpret, and we have now revised these panels accordingly, also taking the reviewer's previous suggestion (2-1) into account. We note that all available experimental data have been included in Fig. 2B (not all variants were measured in high salt/low pH conditions due to technical difficulties). This is now also indicated in the caption of Fig. 2. We believe the current Fig. 2, with the new panel D and modified B is much easier to interpret, and we thank the reviewer for that.

Comment 2-6:

The calculated bare rates for quenching as a function of pH are the same within experimental error, but still almost 20% different. Is it possible to get more accuracy on these measurements? It might be useful to correct the comparisons between the protein PET results.

Improving the uncertainty of CW relaxation times derived from PET experiments is a continuous effort in our lab. We have made every effort to maximize the accuracy of our bimolecular quenching rate measurements, including careful calibration and repeated experiments to minimize variability. The current level of precision reflects the best achievable with our present methodology. We have added a paragraph clarifying how the error is estimated in the Supporting Methods S2. As the reviewer points out, the observed variation falls within experimental uncertainty and therefore does not impact our conclusions. While further reducing this uncertainty remains an ongoing goal in our lab, it would require substantial technical advances beyond the scope of the current study.

Comment 2-7:

The all-atom simulations were (understandably) only done at one pH, and if I follow the logic correctly, the argument is that since the “physiological” PET rates being so very different between C3W2 and C4W2 is recapitulated in the all atom model, then they are relevant. However, the coarse-grained model were also able to reproduce that difference between rates, but unable to capture the changes due to the pH shift. This is all fine, but from a writing perspective, it makes it unclear whether the all atom have really succeeded in reproducing the experimental data. I wonder, simply from a story-telling perspective, if it would better to focus on the increased sequence information given by the all atom rather than the limitations of the coarse grained model.

The reviewer is correct that all-atom simulations accurately capture the dynamic heterogeneity between the C3W2 and C4W2 relaxation times (2.7 ratio) that was observed in measurements (2.8 ratio) and therefore are relevant for investigating the molecular origin of this phenomenon. Although the coarse-grained simulations predicted a dynamic heterogeneity for the two PET constructs, the extent of the heterogeneity (1.6 ratio) was much closer to that of the homopolymer (1.3 ratio) than the measurements.

We believe that the reviewer was confused about this point because in the original version of the manuscript, the all-atom and CG ratios were not quantified in the main text, and the CG ratio was not presented clearly in Fig. 2D. Although the all-atom relaxation times were presented and discussed in Supporting Methods S17 in the original version, the dynamic heterogeneity and C3W2/C4W2 ratios were not explicitly quantified there either. This was a major shortcoming of the original manuscript, and we would like to thank the reviewer again for pointing it out. In the revised manuscript we have changed Fig. 2D, quantified and discussed the ratios in detail in the main text. We have also restructured pages 12-19 to present our results more clearly, and to highlight the strengths and limitations of both CG and all-atom simulations.

Finally, to reinforce our claim that the all-atom simulations did indeed reproduce the experimental data accurately we now have also estimated the other three relaxation times as well (shown in Fig. S30). In addition, we would like to point out that all-atom simulations also reproduced other types of independent experimental data as well, such as CD spectra for the wild-type and W2 variants, as well as SAXS and NMR chemical shifts for wild type N_{TAIL}, which further increased our confidence in the simulations.

Reviewer #3

In this study, the authors explored how certain internal interactions within the measles virus nucleoprotein (NTAIL) affect its behavior and role, especially in its connection with the polymerase complex, which helps the virus replicate and transcribe. They used experimental and computational methods and discovered that short-lived interactions between separate parts of NTAIL are crucial in shaping its behavior and interactions with other proteins, influencing the virus's ability to replicate and transcribe. The findings suggest these types of interactions might also be important in other similar proteins, although this was not directly investigated or substantiated.

Overall the paper provides a thorough and valuable analysis that contributes significantly to our understanding of these proteins and should definitely be published.

We thank the referee for the very positive assessment.

Comment 3-1:

While I have no major concerns about the scientific value or quality of the research, the presentation could be clearer and less technical. I honestly had a hard time following many parts of the text and managing the large number of supplementary figures and tables. I know they are important, but they sometimes made it hard to follow the logic of the paper. In summary, many times I felt that the paper was too technical for a broader chemistry audience journal.

We appreciate the comments of the reviewer regarding the clarity, technicality, and reliance on supplementary information to communicate our points. In fact, after re-reading our original manuscript in light of these comments, we fully agree. We have therefore extensively revised the results and discussion sections following the reviewer's comments, which we feel has improved the readability substantially (see also our summary reply to the Editor above). Specifically, we now include summaries of the supplementary materials, so that the reader does not have to switch back and forth between the main text and supplements to understand the main lines of our reasoning. We also improved the logical flow of the text, and have worked hard to make the text more easily understandable for non-expert readers with a chemistry background.

Comment 3-2:

On the other hand, I missed some information that would be better presented in a table. For greater clarity on the computational work, it would help to include details like the duration of each simulation in a table. This table could also specify the number of replicates and other details that distinguish each simulation. This would make it easier for others to replicate the study and to assess whether the results are based on a single analysis or a robust study with multiple replicates that also considers the statistical significance of the results. I honestly don't know at this point the answer for this last point.

We agree and have now added two supplementary tables (Tables S8 and S9) that provide simulation lengths and replicate numbers. Regarding the robustness of our simulation results, the large error bars on PET estimates and large variation of the secondary structure content across replicates might suggest that our simulations are not fully converged yet. However, sampling in our simulations is robust enough to simultaneously explain or reproduce several of the experimental observations about N_{TAIL} dynamics, by one set of simulations. Further the two most accurate tested force fields, CHARMM36m-OPC and Amber99SB-disp, independently predict a similar dynamic heterogeneity for the C3W2 and C4W2 variants, as well as a free-energy landscape with similar features (data not shown), which also underscores the robustness of our all-atom simulations.

Comment 3-3:

Rearrange Figure S24 to improve visibility. Consider placing part A above part B and enlarging A, as it is quite hard to see the details described on page 35.

Thank you for the suggestion. We resized and adjusted the Figure accordingly.

Comment 3-4 :

In Table S1, please clarify what the colors represent.

Thank you! We have now added the necessary clarification to the Table header.

Reviewer #4:

The manuscript by Otteson et al. the interactions between amino acid residues that are distant in sequence of the N_tail protein of the measles virus are investigated using photo-induced electron transfer (PET) and Molecular Dynamics (MD) simulations.

Comment 4-1:

From detected PETs and analysis of extensive MD simulations, the temporary and short-lived interactions between introduced tryptophan and cysteine pairs can be seen. An estimate of the rate of conversion between states S1-S4 from a Markov model would be beneficial but is not absolutely critical here.

This is interesting since the intrinsically disordered protein (IDP) Ntail is so flexible and showing a multi-dimensional conformational surface, that this information cannot be obtained otherwise.

We thank the reviewer for the positive assessment, and agree that a simple and intuitive kinetic model would be useful to describe the transition rates between the four conformational states of Ntail. In fact, we did try fitting Markov models to the free energy landscape before submission and obtained averaged transition rates between the states. Unfortunately, upon closer inspection, the shallow free-energy landscape of Ntail, as for many other IDPs, renders analysis with Markov models challenging and potentially misleading.

Specifically, we noticed that – at least when using a number of Markov states small enough to facilitate an intuitive understanding – several criteria of Markovianity were violated. In particular, the duration of actual transitions between the states were generally not much shorter than the preceding dwell times. Further, we observed multiple transition modes with different kinetics between the same pairs of states. Addressing these challenges would have either required a large number of Markov states along these transitions or the use of non-Markovian kinetic models including memory effects. In both cases, it is unlikely that our sampling would have sufficed to define the much larger number of parameters of the models. Additionally, both of these solutions would likely yield less intuitive results than a simple Markov model, would have required the description of yet another complex method, and would make maintaining the actual storyline and focus even more challenging.

For these reasons, we decided to only include the free-energy landscape in Fig. 3 in the manuscript, which suffices to support our main conclusions. Nevertheless, and encouraged by the referee's comment, on pages 18-20 of the revised manuscript, we now provide a more in-depth description of the free energy landscape to better explain how non-local interactions give rise to the observed dynamic heterogeneity.

Comment 4-2:

It is clear that the authors extend the system beyond the MoRE region flanked by residues of N_{tail} to either side but we would need to see a discussion of previous work by NMR and all-atom MD (their references 19 and 26). Does this region retain its secondary structure in solution in experiment and solution and how do the results of this work agree with refs 19 and 26.

We thank the reviewer for this request, and have added such a discussion to the Supporting Methods S23 section of the revised manuscript as described below. We would like to stress, however, that while such consistency checks with existing literature are certainly helpful and fair, the main conclusions of our manuscript rest on our direct comparisons to CD and NMR measurements. Here is our summary of previous work, to our best knowledge:

Robustelli *et al.* (ref 26) used the Amber99SB-disp force field to simulate the MoRE region, and found a 20% helix propensity, and good agreement with C α chemical shift (RMSD: 0.52 ppm, and R: 0.92). Wang *et al.* (ref 19) performed temperature replica exchange MD simulations using the CHARMM22* force field, and estimated a similar 20% helix content in the MoRE region, with a slightly lower Pearson Correlation (R: 0.76). Both simulations assigned considerably higher helical propensity (50-60%) in P_{XD} the binding site (491-501) than the preceding segment (482-490).

Our two best wild-type N_{TAIL} simulations were performed using CHARMM36m-OPC force field (17% helix content, RMSD: 0.47 ppm, and R: 0.83) and the Amber99SB-disp force field (18% helix content, RMSD: 0.36 ppm, R: 0.84) provided by Piana *et al.* Both of these forcefields predict a transient helix at the MoRE that both folds and unfolds. The two force fields also agree in that the binding site (491-501) has a higher content than the preceding segment.

As far as we can tell, no NMR data is available for the PET variants, but the measured CD spectra of W2 variant indicate a change in the secondary structure composition compared to the wild type construct (likely with a small decrease of the helix content). Simulation trajectories C3W2 and C4W2 variants using CHARMM36m-OPC show a smaller helix content (~10%) which is more compatible with the W2 CD spectrum. In contrast, Amber99SB-disp simulations predict an increased helix content in the MoRE region (~29%) as well as in the transient helices outside of it. The helix content of MoRE residues in our simulations are summarized in Supplementary Figure S31.

Comment 4.3:

In experiment and MD simulations, the authors show that it is electrostatic interactions between differently positioned amino acid residues, which is responsible for the heteropolymeric behavior of N_{tail}. It remains to be mentioned critically, that the first coarse-grain (MD) simulations required the adjustment of parameters to be able to get the radius of gyration and the helical propensity of the MoRE region close to experimental values from SAXS and NMR.

We fully agree and now clearly state in the revised version of the manuscript (page 10) that the CG models for N_{TAIL} were fitted to reproduce SAXS and NMR data. Coarse-grained models offer computational efficiency, but they often do not capture all sequence-specific features accurately. This is one of the reasons, why we also included atomistic simulations, despite the considerable computational effort.

Comment 4-4:

When altering the pH in experiment and simulations, a lower pH of 4.0 leads to 'more neutral' Asp and Glu residues and neutral His will become 'more positively charged'. This statement is correct but a quantification is needed, how many protons will be added to these residues at pH 4.0. It requires an estimation of the pKa values of the individual residues and then an annotation of change of N_tail residue by pKa.

Indeed, the description in the original manuscript was somewhat vague. In the revised version of manuscript (p. 12-13), it is now clearly stated that pH effects in CG simulations are modeled by partial charges that reflect the average protonation state of protonatable residues under the relevant conditions. Such a simplified approach is suitable at the level of CG modeling, where the goal is to represent overall electrostatic changes rather than detailed titration states. By contrast, all-atom simulations at low pH would indeed require assignment of discrete protonation states or the use of constant-pH molecular dynamics methods, which would certainly be interesting in future work.

Comment 4-5:

On the other hand, on p 13, line 9 the authors state that changes in PET are 'entirely due to changes in the protein, rather than pH-induced changes of the electron transfer efficiency'. It is clear that the first will effect the latter. pH-induced changes in protein structure will critically effect the distances between W and C residues.

We thank the reviewer for pointing that our wording was unclear. We agree that pH-induced structural changes in the protein directly affect C-W distances and thus affect the measured relaxation times due to PET. However, our statement was intended to clarify that the intrinsic electron transfer efficiency (at contact) is not significantly altered by pH. As shown in Figure S19, the bimolecular quenching rate between W and C freely diffusing in solution, varies only slightly across different pH conditions, indicating that changes in PET rates observed in the constructs are primarily due to structural rearrangements rather than shifts in the intrinsic quenching efficiency. We have now revised the text to make this distinction clearer.

Comment 4-6:

In Figure 4, the side chain salt bridge and C_alpha backbone interactions are shown as contact maps. These are very hard to see and a better coloring scheme or radius is required.

We agree that the individual interactions are difficult see in Fig. 4. Although we note that the primary function of the figure is to highlight the main interaction clusters that stabilize the four conformational states and the differences between them, we have now improved the visibility of individual salt bridges and sporadic interactions. Specifically, we changed the color scheme of the contact maps and applied an interpolation method to increase salt-bridge pixel size. In addition, we provided the stand-alone contact maps at higher resolution for each state in the supplementary materials (Supplementary Figs. S23-S26, without interpolation).

Comment 4-7:

On p 18, line 3, the authors claim to have validated the choice of force field for the protein and water against experimental CD and PET experiments. However, we see only two combinations CHARMM36m/OPC and AMBER99SB-disp/TIP4P in the methods part. The authors need to clarify this.

We would like to clarify that we did the validation in two steps, which may not have been apparent in the original version of the manuscript, as most of the discussion was relegated to the Supporting Methods S10. In the first step we used 8 different force fields to simulate WT N_{TAIL}, for which extensive experimental data were available. In the second step, we used the resulting two best performing forcefields and performed simulations on the two PET variants (C3W2 and C4W2) to address the dynamic heterogeneity. The PET calculations were performed only for the PET variant simulations, as accurate PET predictions require a tryptophan residue at the appropriate sequence position to capture the short-range contact dynamics of C-W pairs. For this reason, we only computed and presented PET comparisons for the best two forcefields. To avoid this potential confusion, we have now revised our description of the forcefield validation, and highlighted the outlined points on page 17.

Comment 4-8:

In Figure 5, the dashed lines are not visible and the figure has to be improved in clarity.

Thank you! We made the dashed lines more visible in the Figure, streamlined the figure contents, and added clarification to the caption.

Comment 4-9:

The conclusion section spans from pages 26 to 36 and appears before the summary. First, the summary ought to be communicated to the reader and then a SHORT section of conclusions can be drawn to speculate about the results of this work on isolated N_{tail} with relevance to its physiological process. However, the latter is tentative only and should be presented in a concise way.

We agree with the reviewer that our manuscript organization was confusing in this regard. In the revised manuscript we therefore restructured the main text as suggested. The Summary was renamed to Conclusions (which is now shorter than one page), and the previous “Conclusions” section has been more fittingly renamed “Discussion”. In the process, we have also shortened it by two pages.

Although tentative on its own, we believe that our discussion about the physiological aspects of our results is an important part of the manuscript that is supported by a body of evidence available in the literature. Two of four reviewers specifically highlighted this part and appear to have shared this belief. In line with the referee’s remarks, we now present this part more concisely without loss of information.

Comment 4-10: (Summary)

Overall, the study is interesting and demonstrates the complexity of the conformational dynamics and flexibility of intrinsically disordered proteins and their possible function in biological processes. Rationalizing PET experiments for two pairs of mutated C/W residues requires VERY extensive sampling by classical MD simulations.

We thank this Reviewer for the overall positive assessment and for the many suggestions which helped us to substantially improve our manuscript.